# DP-GPL: Differentially Private Graph Prompt Learning

## Abstract

Graph Neural Networks (GNNs) have shown remarkable performance in various
applications. Recently, *graph prompt learning* has emerged as a powerful GNN
training paradigm, inspired by advances in language and vision models. Here,
a GNN is pre-trained on public data and then adapted to sensitive tasks using
lightweight graph prompts. However, using prompts from sensitive data poses
privacy risks. In this work, we are the first to investigate these risks in graph
prompts by instantiating a membership inference attack that reveals significant
privacy leakage. We also find that the standard privacy method, DP-SGD, fails to
provide practical privacy-utility trade-offs in graph prompt learning, likely due to
the small number of sensitive data points used to learn the prompts. As a solution,
we propose two algorithms, `DP-GPL` and `DP-GPL+W`, for differentially private
graph prompt learning based on the PATE framework, that generate a graph prompt
with differential privacy guarantees. Our evaluation across various graph prompt
learning methods, GNN architectures, and pre-training strategies demonstrates that
our algorithms achieve high utility at strong privacy, effectively mitigating privacy
concerns while preserving the powerful capabilities of prompted GNNs.

## 1 Introduction

Graph Neural Networks (GNNs) have emerged as a powerful tool for learning representations of
graph-structured data and have shown significant advancements across various applications, such
as drug design (Al-Rabeah & Lakizadeh, 2022; Qian et al., 2023), anomaly detection (Sun et al.,
2022b; Tang et al., 2022), and social network analysis (Chen et al., 2020). Recently, *graph prompt
learning* (Sun et al., 2023d; Zi et al., 2024; Sun et al., 2023b; Fang et al., 2024; Sun et al., 2022a;
2023a) has emerged as a promising GNN training paradigm. Graph prompt learning first pre-trains a
GNN model on general public graph data and then tunes a graph prompt (Sun et al., 2023b; Huang
et al., 2024; Ge et al., 2023) or tokens (Fang et al., 2024; Sun et al., 2022a; Liu et al., 2023b) on
some sensitive downstream data. By reformulating the downstream task into the pretext task used in
pre-training, it then enables predictions for the downstream task.

The fact that graph prompts are tuned on sensitive downstream data can raise significant privacy
concerns. In fact, in the language and vision domains, it has been shown that private information
from downstream data can leak through predictions of prompted models (Duan et al., 2023b; Wu
et al., 2023). To the best of our knowledge, no such insights exist for the graph domain, and no prior
work has explored the privacy risks of graph prompt learning.

In this work, we set out to close this gap. We first assess the privacy risks of graph prompts by
adapting a state-of-the-art membership inference attack (Shokri et al., 2017; Carlini et al., 2022) to
graph prompt learning and measuring the empirical leakage. Our evaluation demonstrates significant
privacy risks for the downstream data when used to tune graph prompts. For example, we show that
the membership inference attack can achieve an AUC score as high as 0.91 on the PubMed dataset.
We also investigate the relationship between the number of data points used to tune the prompt and the
attack success and find that with less data, the privacy risk grows, posing a significant risk to standard
graph prompt learning that usually relies on a small number of data points (Sun et al., 2023a).

As a naive solution to mitigate this privacy risk, we first turn to the Differential Privacy-Stochastic
Gradient Descent (DP-SGD) algorithm (Abadi et al., 2016)—a gold standard in privacy-preserving
machine learning. However, we find that this approach significantly degrades the downstream

Figure 1: Framework of DP-GPL+W. ❶ We partition the labeled private data into disjoint groups according to the centrality score of each node. ❷ An ensemble of teacher prompts is trained on the disjoint private data groups. ❸ Given an unlabeled public data sample, by querying the pre-trained GNN model, each teacher prompt votes with the most confident class label. ❹ According to the average centrality score of each private data group, the teacher prompts' votes are weighted aggregated, *i.e.*,, the higher the centrality score, the more weight the teacher prompt's vote has. A noisy argmax over weighted vote counts is returned as the final noisy label for the public data sample. ❺ A student prompt is trained with the labeled public data and can be publicly released.

performance due to the limited amount of data used to tune graph prompts. For instance, with a privacy budget as high as $\varepsilon = 64$, the accuracy on the Cora dataset downstream drops from 48.70% to 18.47%, *i.e.,* close to random guessing.

As a solution for practical privacy-preserving graph prompt learning, we propose two new algorithms, DP-GPL and its variant DP-GPL+W. DP-GPL follows the general framework of the private aggregation of teacher ensembles (PATE) (Papernot et al., 2017; 2018), but instead of training a student model with differential privacy guarantees, it trains a student prompt (Duan et al., 2023a). DP-GPL+W additionally leverages the inherent structure of the graph data and the insight that different nodes in a graph may have different influence levels. Based on these, it integrates a novel data partitioning algorithm for the teacher prompts to improve privacy-utility trade-offs further. Concretely, DP-GPL+W groups nodes with respect to their centrality score, assigns nodes with similar levels of centrality to the same teacher, and during the voting process, weights the teachers' votes according to their nodes' centrality (*i.e.,* influence). We thoroughly evaluate our algorithms in terms of privacy guarantees and privacy-utility trade-offs. Over various graph prompt learning methods, GNN architectures, and pre-training strategies, we find that our algorithms achieve high utility at strong privacy privacy guarantees—thereby, implementing the first practical approach to private graph prompt learning.

In summary, we make the following contributions:

- We are the first to show that private information can leak from graph prompts, in particular when the prompts are tuned over a small number of data points.
- We show that naively integrating the DP-SGD algorithms into graph prompt learning yields impractical privacy-utility trade-offs.
- As a solution, we propose DP-GPL and DP-GPL+W, two algorithms based on the PATE framework to implement differential privacy guarantees into graph prompt learning.
- We perform a thorough evaluation on multiple state-of-the-art graph prompt learning methods, graph datasets, GNN models, and pre-training strategies and highlight that our new methods achieve both high utility and strong privacy protections over various setups.

## 2 BACKGROUND AND RELATED WORK

### 2.1 PROMPT LEARNING

Prompt learning is a new machine learning paradigm that has been recently proposed to improve the performance of large models while addressing the limitations of fine-tuning (Li & Liang, 2021; Lester

et al., 2021; Liu et al., 2023a). The idea is to learn a task-specific prompt that can be added to the input data while freezing the pre-trained model's parameters. In addition to many effective prompt methods in the language domain, such as hand-crafted textual prompts (Brown, 2020), automated discrete prompts (Gao et al., 2020; Shin et al., 2020), and trainable prompts in the continuous space (Li & Liang, 2021; Liu et al., 2021), also in the vision domain (Jia et al., 2022; Sohn et al., 2023) and for multi-modal models (Zhou et al., 2022), prompt tuning has become a prevalent paradigm.

## 2.2 GNNs and Graph Prompt Learning

GNNs achieve strong performance on numerous applications (Sun et al., 2023c; Tang et al., 2022; Chen et al., 2020). Therefore, they rely on various effective architectures, such as Graph Convolutional Network (GCN) (Kipf & Welling, 2022), Graph Attention Network (GAT) (Veličković et al., 2018a), and Graph Transformer (Shi et al., 2020)—usually trained in a *supervised* manner. To make graph learning more adaptive, many graph pre-training approaches have been proposed (Veličković et al., 2018b; Hou et al., 2022; Sun et al., 2022a; Xia et al., 2022) that first learn some general knowledge for the graph model with easily accessible data, and then fine-tune the model on new tasks. This is often referred to as *"pre-train & fine-tune"* paradigm. However, the large diversity between graph tasks with node level, edge level, and graph level may cause a "negative transfer" results where the knowledge learned during the pre-training phase hurts performance when fine-tuning on a specific downstream task, rather than improving it (Sun et al., 2023b). As a solution, graph prompt learning was proposed. The goal of graph prompt learning is to learn transformation operations for graphs to reformulate the downstream task to the pre-training task. It can be formulated as follows:

$$\Phi(\mathcal{P}(X, A, X^*, A_{inner}, A_{insert})) = \Phi(\mathcal{T}(X, A)) \tag{1}$$

where $\Phi$ is the frozen pre-trained graph model, $X \in \mathbb{R}^{N \times d}$ and $A \in \{0, 1\}^{N \times N}$ are node feature matrix and adjacency matrix of the original graph $\mathcal{G}$ respectively. $\mathcal{P}$ is a graph prompt learning module that learns the representations of K prompt tokens, *i.e.,* $X^* \in \mathbb{R}^{K \times d}$, token structures, *i.e.,* $A_{inner}$ and inserting patterns, *i.e.,* $A_{insert}$, which indicates the connection between the prompt tokens and the nodes in the original graph. $\mathcal{T}$ indicates any graph-level transformation, showing that we can learn a graph prompt learning module $\mathcal{P}$ applied to the original graph to imitate any graph-level transformation. While Equation (1) shows graph-level transformation, our adaption of graph prompt is in node-level, *i.e.,* the graph prompt is learned only based on the selected nodes' features without the adjacency matrix $A$ of the original graph $\mathcal{G}$. In addition, the learned graph prompt is adapted to individual nodes, *i.e.,* $\mathcal{P}(x)$ where $x$ is an individual node.

For instance, Graph Pre-training and Prompt Tuning (GPPT) (Sun et al., 2022a) applies prompt-based tuning methods to models pre-trained by edge prediction. It introduces virtual class-prototype nodes/graphs with learnable links into the original graph, making the adaptation process more akin to edge prediction. Fang et al. (2024) proposed a universal prompt-based tuning method, called Graph Prompt Feature (GPF), which can be applied under any pre-training strategy. GPF adds a shared learnable vector to all node features in the graph while its variant GPF-plus incorporates different prompted features for different nodes in the graph. Sun et al. (2023b) proposed All-in-one, a graph prompt that unifies the prompt format in the language area and graph area with the prompt token, token structure, and inserting pattern. They reformulate the downstream problems to the graph-level task to further narrow the gap between various graph tasks and pre-training strategies. Graph prompt learning has superior performance compared to traditional fine-tuning methods and is especially effective in few-shot settings, *i.e.,* when only a small number of data points are sampled to tune the prompt. While graph prompt learning benefits various graph applications, in this work, we focus on node classification tasks and three state-of-the-art graph prompt learning methods, namely GPPT, All-in-one, and GPF-plus.

## 2.3 Privacy Risks in GNNs and Graph Prompt Learning

GNNs have been shown to be vulnerable to various privacy risks, such as membership inference attacks (MIAs) (Olatunji et al., 2021; He et al., 2021; Conti et al., 2022), model inversion attacks (Zhang et al., 2022a), and property inference attacks (Wang & Wang, 2022; Zhang et al., 2022b). Specifically, MIAs against GNNs aim to infer whether a given node or graph was used to train the GNN model, model inversion attacks aim to recover the model's training data from the model's output, and property inference attacks aim to infer the sensitive properties of the training data through

the access to the target GNN model. Regarding graph prompt learning, some prior work explores backdoor attacks in graph prompt learning, which utilize prompts to insert backdoor triggers into the GNN model (Lyu et al., 2024) to impact output integrity. To the best of our knowledge, there is no prior work on assessing and mitigating the privacy risks in graph prompt learning.

## 2.4 DIFFERENTIAL PRIVACY

Differential privacy (DP) (Dwork, 2006) is a mathematical framework that provides privacy guarantees for randomized mechanisms $\mathcal{M} : I \to S$. Therefore, it upper-bounds the probability that $\mathcal{M}$, when executed on two neighboring datasets $D$, $D'$, *i.e.,* dataset that differ in only one data point, output a different result by formalizing that $\Pr[\mathcal{M}(D) \in S] \le e^{\epsilon} \cdot \Pr[\mathcal{M}(D') \in S] + \delta$. The privacy parameter $\varepsilon$ specifies by how much the output is allowed to differ, and $\delta$ is the probability of failure to meet that guarantee. There are two main algorithms to implement DP guarantees for traditional machine learning. The **differentially private stochastic gradient descent** algorithm (DP-SGD) (Abadi et al., 2016) extends standard stochastic gradient descent with two additional operations, first, gradient clipping that limits the impact of each individual training data point (often called *"sensitivity"*) on the model update, and then the addition of calibrated amounts of stochastic noise to provide formal privacy guarantees. The second **private aggregation of teacher ensembles** algorithm (PATE) (Papernot et al., 2017; 2018) trains an ensemble of *teacher* models on disjoint subsets of the private data. Then, through a noisy labeling process, the ensemble privately transfers its knowledge to an unlabeled public dataset. Finally, a separate *student* model is trained on this labeled public dataset for release.

**DP for Graphs.** As the classical DP guarantee makes no assumptions about potential correlations between data points, there are existing works that extend DP on graph data (Mueller et al., 2024; Sajadmanesh et al., 2023; Kasiviswanathan et al., 2013; Olatunji et al., 2023; Sajadmanesh & Gatica-Perez, 2024; Xiang et al., 2024). There are three variants of DP on graph data: node-level DP, edge-level DP, and graph-level DP, depending on what the data owner requires to protect. Specifically, node-level DP aims to protect the privacy of individual nodes in the graph data, including its attributes and associated edges (Sajadmanesh et al., 2023; Kasiviswanathan et al., 2013; Daigavane et al., 2021; Olatunji et al., 2023). Edge-level DP aims to protect the relationships between nodes, which can be applied to social network graphs (Hay et al., 2009) or location graphs (Xie et al., 2016), where the edges contain sensitive information, but the data represented in the nodes of the graph are assumed to be non-sensitive. Graph-level DP aims to protect the entire graph data, including the structure of the graph, node attributes, and edge relationships (Mueller et al., 2022). However, graph-level DP has not been thoroughly investigated in the literature (Mueller et al., 2024). In this work, we focus on *node-level DP* as we aim to protect the privacy of individual nodes in the graph data. Different from the existing node-level DP guarantees (Sajadmanesh et al., 2023; Kasiviswanathan et al., 2013; Olatunji et al., 2023; Sajadmanesh & Gatica-Perez, 2024; Xiang et al., 2024), which often results in large $\epsilon$ values, limiting their practical utility, we aim to achieve meaningful privacy guarantees for graph prompt learning with small and manageable $\epsilon$ values ($\epsilon <= 2$).

## 2.5 PRIVATE PROMPT LEARNING IN THE VISION AND LANGUAGE DOMAIN

In the vision domain, Li et al. (2023) leverage the PATE algorithm for private prompt tuning to vision encoders. Therefore, they have to tune a prompt and train an additional label mapping for each teacher. In contrast, our method instantiates different teachers only through graph input prompts. In the language domain, multiple approaches have been proposed to privatize prompts. Chen et al. (2023) rely on named entity recognition to identify and hide private information in text prompts. This approach is not easily transferable to the graph domain and additionally does not yield formal privacy guarantees. The DP-OPT (Hong et al., 2024) framework relies on a local large language model (LLM) to derive a discrete, *i.e.,* text, prompt with DP, and then transfers this prompt to a central LLM. The framework is tightly coupled to the language domain and derives plain language prompt templates that are not applicable to GNNs. Panda et al. (2023) rely on a PATE-style teacher ensemble implemented through different prompts, and generate noisy output predictions for the LLM. Yet, due to the absence of a student model in their framework, each query to the ensemble consumes additional privacy budget, making the approach impractical. Duan et al. (2023a) solve this limitation by generating a student prompt from the teacher ensemble, similar to our work. Yet, they treat all

teachers in the ensemble equally, which can yield sub-optimal privacy-utility trade-offs. In contrast, we leverage the inherent structure of the graph data to identify more important data points and weight their teachers' votes higher, improving privacy-utility trade-offs.

# 3 PRIVACY RISKS IN GRAPH PROMPT LEARNING

In this work, we explore the privacy risk for the sensitive downstream data in graph prompt learning by instantiating a MIA(Carlini et al., 2022; Shokri et al., 2017). While prior work on instantiating MIAs against natural language prompts relies on a simple threshold-based attack (Duan et al., 2023b), we adapt and implement the more powerful state-of-the-art Likelihood Ratio Attack (LiRA) (Carlini et al., 2022).

We use this attack to assess whether a given data point was used to train a given target prompt. Formally, in our MIA, we consider that the goal of the adversary is to infer whether a given private data sample $v = (x_p, y_p)$ is in the training dataset of the target prompt $\mathcal{P}_{target}$. We assume that the adversary holds $n$ candidate nodes $(x_1, x_2, \ldots, x_n)$ including their corresponding labels $(y_1, y_2, \ldots, y_n)$ and queries the candidates nodes with prepended target prompt to the pre-trained GNN model.

The pre-trained GNN model then outputs the probability vectors $(p_1, p_2, \ldots, p_n)$. Following Carlini et al. (2022), we analyze the model's output probability at the correct target class label of every candidate node $x_i$, *i.e.*, $p_{i,y_i}$. The intuition of this MIA is that the output probability at the correct class $y_i$ will be significantly higher for members that were used in training $\mathcal{P}_{target}$ than non-members. The detail of our adaptation of the LiRA attack to the graph prompt learning setup is presented in Algorithm 1.

---

**Algorithm 1 Likelihood Ratio Attack on Graph Prompt Learning.** Instead of conducting MIA against the target model in the standard LiRA algorithm, we conduct MIA against the target prompt in graph prompt learning. We highlight these differences in blue.

---

**Require:** Target prompt $\mathcal{P}_{target}$, Pre-trained GNN model $\Phi$, A given data sample $(x_p, y_p)$, data distribution $\mathbb{D}$, Logit scaling $f(p) = log(\frac{p}{1-p})$
1: $\text{confs}_{in} = \{\}$, $\text{confs}_{out} = \{\}$
2: **for** $i \leftarrow 1$ to $K$ times **do**
3:     /* Sample a shadow dataset */
4:     $D_{attack} \leftarrow^{\$} \mathbb{D}$
5:     /* Train IN graph prompt */
6:     $\mathcal{P}_{in} \leftarrow \mathcal{T}(D_{attack} \cup (x_p, y_p))$
7:     $\text{confs}_{in} \leftarrow \text{confs}_{in} \cup \big\{ f(\Phi(\mathcal{P}_{in}(x_p))_{y_p}) \big\}$
8:     /* Train OUT graph prompt */
9:     $\mathcal{P}_{out} \leftarrow \mathcal{T}(D_{attack} \setminus (x_p, y_p))$
10:     $\text{confs}_{out} \leftarrow \text{confs}_{out} \cup \big\{ f(\Phi(\mathcal{P}_{out}(x_p))_{y_p}) \big\}$
11: **end for**
12: $\mu_{in} \leftarrow \text{mean}(\text{confs}_{in})$, $\mu_{out} \leftarrow \text{mean}(\text{confs}_{out})$
13: $\sigma_{in}^2 \leftarrow \text{var}(\text{confs}_{in})$, $\sigma_{out}^2 \leftarrow \text{var}(\text{confs}_{out})$
14: /* Query with target graph prompt */
15: $\text{conf}_{obs} = f(\Phi(\mathcal{P}_{target}(x_p))_{y_p})$
**Ensure:** $\Lambda = \frac{p(conf_{obs}|\mathcal{N}(\mu_{in}, \sigma_{in}^2))}{p(conf_{obs}|\mathcal{N}(\mu_{out}, \sigma_{out}^2))}$

---

**MIA Experimental Setup**. We conduct MIA against graph prompt learning on three downstream datasets, *i.e.*, Cora, CiteSeer, and PubMed with GNN models pre-trained on the ogbn-arxiv dataset.[1] To evaluate our MIAs under different numbers of data points used to tune the graph prompt, we analyze MIA in 1-5 shot settings. Following the experimental setup from MIAs against natural language prompts (Duan et al., 2023b), for each experiment, we consider the $k$ (*i.e.*, $k$=1-5) data points used in training the target prompt as members and $50 * k$ other randomly selected data points from the testing dataset as non-members. We repeat the MIA 100 times and report the average attack success.

**MIA Results**. In Figure 2, we present the AUC-ROC curve of our MIA on the Cora dataset and the GAT model. The results for other datasets and models are

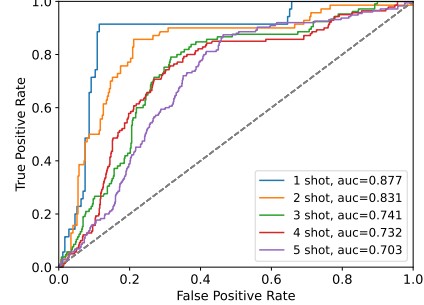

Figure 2: **AUC-ROC curve of our MIA on Cora dataset with different number of shots, *i.e.*, 1-5 shots.** With fewer shots, MIA success rises significantly.

---

[1] Details of these datasets are presented in Section 5.1.

presented in Appendix A.4.2 and show a similar trend.

Our results highlight that the privacy risk increases with fewer shots used to train the prompt, *e.g.,* with 5 shots we have an AUC score of 0.703, while with 1 shot, the AUC score increases to 0.877. We hypothesize that this is due to the fact that with fewer shots, the target prompt is more likely to overfit the prompt data, leading to a higher membership inference risk. Yet, even with more shots, we observe significantly higher MIA success than the random guessing (0.5), *e.g.,* see Figure 3 with 5-shots over various setups where the average AUC score is consistently between 0.7-0.9. Hence, our results demonstrate that the private data used in training a graph prompt can be subject to substantial privacy risk. This motivates the urgent need for privacy-preserving graph prompt learning methods.

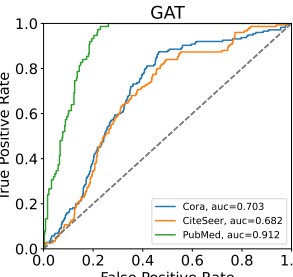 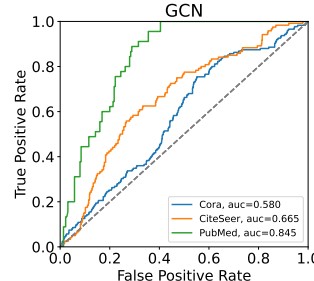 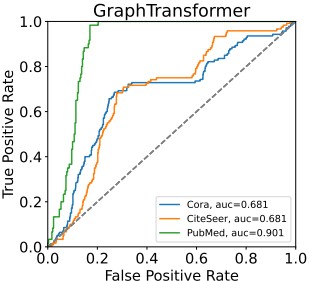

Figure 3: **AUC-ROC curve of our MIA (with 5 shots).** Generally, there is a high MIA risk in terms of AUC score of between 0.7-0.9.

## 4 TOWARDS PRIVACY PRESERVING GRAPH PROMPTS

The standard approach for privacy-preserving machine learning is based on the DP-SGD algorithm (Abadi et al., 2016). The DP-SGD algorithm can be applied in gradient-based learning approaches to limit the impact of individual training data points on the final model and add calibrated noise to implement the privacy guarantees. We explore this naive way of implementing privacy guarantees into graph prompt learning and show that it fails to yield reasonable utility even at low privacy regimes, *i.e.,* with very high $\varepsilon$'s. Motivated by this insight, we propose two non-gradient based algorithms for private graph prompt learning based on the PATE framework.

### 4.1 NAIVE IMPLEMENTATIONS OF PRIVACY IN GRAPH PROMPT LEARNING FAIL

As a naive solution to yield private graph prompt learning, we rely on the DP-SGD algorithm. Therefore, we keep the GNN frozen, calculate the gradients only with respect to the graph prompts, clip and noise them according to the desired privacy protection, and update the prompt iteratively to minimize the loss on the downstream task. Our evaluation of this naive approach in Table 6 in Appendix A.4.3 highlights that DP-SGD yields inadequate privacy-utility trade-offs for private graph prompt learning. While our results show the general trend that with increasing privacy budgets, the performance of the downstream task increases, DP-SGD still significantly degrades the downstream task performance even at high privacy budgets. For instance, with a privacy budget as high as $\varepsilon = 64$ in the 5-shot setting, the accuracy of the downstream task on the Cora dataset still drops from 48.70% to 18.47%, which is close to random guessing.

### 4.2 TWO DIFFERENTIALLY PRIVATE GRAPH PROMPT LEARNING FRAMEWORKS

Motivated by the failure of the naive DP-SGD approach, we propose two non-gradient based DP graph prompt learning frameworks, `DP-GPL` and its variant `DP-GPL+W`. We detail the general workflow of `DP-GPL+W` in Figure 1. Note that `DP-GPL` follows a similar structure but differs in the *private data partitioning* and *aggregation* blocks as detailed below.

Following PATE (Papernot et al., 2017; 2018), our algorithms contain the broader stages of training the teacher models, performing a private knowledge transfer, and obtaining the student. In contrast to standard PATE, we do not train teachers from scratch, but using the same frozen pre-trained GNN, we tune teacher prompts. Additionally, our student is again not a trained model like in PATE, but a

prompt tuned on the public data labeled during the knowledge transfer. As an additional difference, `DP-GPL+W` does not perform the data partitioning at random, as done in PATE. Instead, it groups nodes according to their centrality score and assigns them to teachers accordingly. We detail the building blocks of our `DP-GPL` and `DP-GPL+W` below:

**Private Data Partition**. In `DP-GPL`, we partition and assign the private data to the teachers at random, the same as PATE. In contrast, in `DP-GPL+W`, we calculate the centrality score of each node in the private dataset, *i.e.*, $c_i = deg(v_i)/(n-1)$, where $deg(v_i)$ is the degree (number of edges) of node $v_i$ and $n$ is the number of nodes in the private graph. Then, we partition the full set of private data points into disjoint groups according to these centrality scores, *i.e.*, $\mathcal{G} = \{g_1, g_2, \ldots, g_N\}$, where $N$ is the number of groups, set according to the desired number of teachers. Then, we assign the groups to the different teachers and calculate the weight for each teacher based on the average centrality score of the nodes in its group, *i.e.*, $\mathcal{S} = \{s_1, s_2, \ldots, s_N\}$. Note that the weight calculation is a one-time operation performed before the training. We make sure that weights stay in a pre-defined range of $[w_{\min}, w_{\max}]$ with $w_{\min}$ and $w_{\max}$ being two hyperparameters that

---

**Algorithm 2** Teacher Weight Calculation for `DP-GPL+W`. Done once during data partitioning.

**Require:** Average centrality scores $\mathcal{S} = [s_1, s_2, \ldots, s_N]$, where $s_i \in [0, 1]$, $w_{\min}, w_{\max}$
**Ensure:** $\mathcal{W} = [w_1, w_2, \ldots, w_N]$, where $w_i \in [w_{\min}, w_{\max}]$ for all $i$, and $\sum_{i=1}^{N} w_i = N$
1: **for** $i = 1$ to $N$ **do**
2:     /* Scale the centrality scores to the range $[w_{\min}, w_{\max}]$ */
3:     $w_i \leftarrow w_{\min} + (w_{\max} - w_{\min}) \times s_i$
4: **end for**
5: /* Normalize the weights to ensure their sum equals $N$ */
6: $S \leftarrow \sum_{i=1}^{N} w_i$
7: **for** $i = 1$ to $N$ **do**
8:     $w_i \leftarrow w_i \times \frac{N}{S}$
9: **end for**
10: **return** $\mathcal{W}$

---

specify the tolerated variation in privacy spending between the teachers. Additionally, we normalize the weights to sum up to $N$, such that we do not need to adjust the noise scale added to privatize the teacher votes from `DP-GPL`. We detail the approach in Algorithm 2.

**Teacher Prompt Tuning.** In this stage, which is alike for `DP-GPL` and `DP-GPL+W`, we tune the teacher prompts according to the data points that were assigned to them. The teacher *prompt* tuning differs from PATE which trains teacher *models* from scratch.

**Public Querying.** To label the public data based on the teacher ensemble, we infer it through the prompted GNN. Therefore, for each teacher, we need to first insert the teacher prompt into the public data. How this insertion is done differs among different graph prompt learning methods. For example, in the GPF-plus method, we insert the teacher prompt into the node features of the public data samples, while in the All-in-one method, the teacher prompt is inserted into the public data as an extra subgraph. Then, we query the pre-trained GNN model once per teacher. For each teacher, we take as a vote the class label with the highest confidence.

**Noisy Teacher Vote Aggregation.** In `DP-GPL`, we assume that each teacher has the same weight, and we aggregate the teachers' votes with a simple majority voting mechanism akin to PATE. In contrast, our `DP-GPL+W` uses the weighted aggregation mechanism illustrated in Figure 4. In the weighted aggregation, we scale every teacher's vote in the histogram according to the teacher's weight from Algorithm 2. Specifically, for a query $\mathcal{Q}$ from the downstream task and classes 1 to $C$, let $y_i(\mathcal{Q}) \in [1, C]$ denote the pre-trained GNN model's prediction for $i$-th teacher prompt, and $c_m(\mathcal{Q})$ denote the vote count for class $m$, *i.e.*, $c_m(\mathcal{Q}) = \sum_i^N (y_i(\mathcal{Q}) = m)$. With vote weight for $i$-th teacher prompt $w_i$, we can get the weighted vote count for class $m$ as follows: $\hat{c}_m(\mathcal{Q}) = \sum_i^N w_i \cdot (y_i(\mathcal{Q}) = m)$. Finally, we add independent Gaussian noise to the weighted count for each class, following the Confident GNMax algorithm (Papernot et al., 2018), and return the label with the highest noisy count for the query.

**Student Prompt Training**. Instead of training a student *model*, like in the original PATE, we use the labeled public data from the aggregation stage to train a *student graph prompt*. This prompt can be released to the public while protecting the private data used to train the teacher prompts.

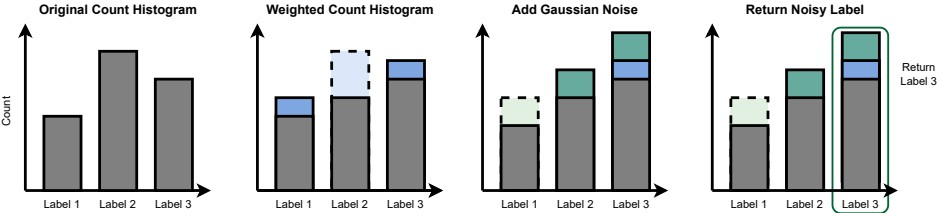

Figure 4: **Weighted Aggregation.** An overview of the weighted aggregation stage in DP-GPL+W. We first turn the standard vote histogram into a weighted histogram based on the teacher weights. Then, we add Gaussian noise to the weighted votes and return the vote with the highest noisy count as the returned label for the public data sample.

### 4.3 PRIVACY ANALYSIS

As the training nodes for different teacher prompts are independent and do not have connecting edges, the privacy analysis of our methods follows that in the original PATE algorithm (Papernot et al., 2018). We analyze the privacy analysis of DP-GPL and DP-GPL+W below.

**DP-GPL.** The privacy analysis of DP-GPL follows the standard privacy analysis of the GNMax algorithm, see Papernot et al. (2018), Section 4.1. Let $f(x)$ denote the histogram obtained by the teacher votes. We use the Gaussian mechanism (Dwork et al., 2014) to obtain a noisy histogram $f'(x)$ as $f'(x) = f(x) + \mathcal{N}(0, \sigma^2)$. We denote by $\Delta_f$ the sensitivity of $f$.[2] The Gaussian mechanism then yields the following data independent bound for PATE (Mironov, 2017):

$$(\alpha, \Delta_f^2 \cdot \alpha/2\sigma^2)\text{-Rényi-DP.} \tag{2}$$

Using standard conversion (Mironov, 2017), we can convert this bound back to $(\varepsilon, \delta)$-DP bounds.

**DP-GPL+W.** The analysis of our DP-GPL+W is significantly more complex due to the different teacher-weighting. In fact, the different weighting of teachers causes them to experience different privacy losses. Hence, instead of yielding homogeneous DP guarantees over all training data points, the algorithm yields heterogeneous DP guarantees (Alaggan et al., 2016), *i.e.*, $(\varepsilon_1, \ldots, \varepsilon_N, \delta)$-DP, with each teacher $i \in N$ and its corresponding prompt data points having $(\varepsilon_i, \delta)$-DP guarantees.

Intuitively, each teacher's privacy loss depends on its weight. A teacher with higher weight can change the voting more, and hence, has a higher sensitivity. In fact, given that the weight is multiplied with the teacher's vote (*i.e.,* 1), the teacher's weight is equivalent to its sensitivity.

**Proposition 1** *A teacher's weight is equal to its sensitivity, i.e., $\Delta_{f,i} = w_i$.*

This proposition leads to the following theorem:

**Theorem 1** *Each teacher $i$ in our DP-GPL+W has a data independent privacy bound of $(\alpha, \Delta_{f,i}^2 \cdot \alpha/2\sigma^2) = (\alpha, w_{f,i}^2 \cdot \alpha/2\sigma^2)$-Rényi-DP.*

*Proof:* The proof follows immediately from replacing $\Delta_f$ in Equation (2) with the correct per-teacher sensitivity $\Delta_{f,i}$. □

After concersion back to $(\varepsilon, \delta)$-DP, DP-GPL+W yields $(\varepsilon_1, \ldots, \varepsilon_N, \delta)$-DP privacy guarantees over all teachers. Each data point from the downstream dataset, hence, obtains the privacy guarantee obtained by the teacher that it is assigned to. Note that while our privacy analysis uses tools and notation from heterogeneous DP (Alaggan et al., 2016), we do not operate in an individualized privacy setup where individual nodes have different privacy requirements. In our setup, all nodes have the same privacy requirement $\varepsilon_{\max}$, and every $\varepsilon_i < \varepsilon_{\max}$ for $\varepsilon_i \in \{\varepsilon_1, \ldots, \varepsilon_N\}$.

---

[2]Given that each teacher can contribute 1 vote, $\Delta_f = 1$ in DP-GPL. We still state the sensitivity explicitly for completeness and as a foundation of the privacy analysis of DP-GPL+W.

## 5 EMPIRICAL EVALUATION

### 5.1 GENERAL EXPERIMENTAL SETUP

**Datasets.** We use ogbn-arxiv (Hu et al., 2020), which is a large-scale graph dataset, as the pre-training dataset. For the downstream tasks, we use Cora (Yang et al., 2016), CiteSeer (Yang et al., 2016), and PubMed (Yang et al., 2016). Since the pre-trained dataset (*i.e.,* ogbn-arxiv) and downstream dataset (*i.e.,* Cora, CiteSeer, and PubMed) have various input feature dimensions, we here use SVD (Singular Value Decomposition) to unify input features from all dimensions as 100 dimensions, following the process in Sun et al. (2023b). We provide more details about these datasets in Appendix A.1. For each dataset, We randomly select 50% of the nodes as the private data and the remaining 50% as the public data. Within the public data, we randomly select 50 nodes as the query nodes and the remaining nodes as the testing data.

**Models.** We use three widely-used GNN models, *i.e.,* **GCN** (Kipf & Welling, 2022), **GAT** (Veličković et al., 2018a), and **Graph Transformer** (GT) (Shi et al., 2020) as the backbone for both "pre-train & fine-tune" and graph prompt learning paradigms. The default hyperparameters used for pre-training GNN models are presented in Table 3. For pre-training strategies, we select four mostly used methods covering node-level, edge-level, and graph-level strategies, *i.e.,* DGI (Veličković et al., 2018b), GraphMAE (Hou et al., 2022), EdgePreGPPT (Sun et al., 2022a), and SimGRACE (Xia et al., 2022).

**Graph Prompt Learning Methods.** Current popular graph prompt learning methods can be classified into two types, 'Prompt as graph' and 'Prompt as token' (Zi et al., 2024). For 'Prompt as graph' type, we select **All-in-one** (Sun et al., 2023b), and for 'Prompt as token' type, we use **GPPT** (Sun et al., 2022a), and **GPF-plus** (Fang et al., 2024). These graph prompt methods are all state-of-the-art. Also, we focus on the 5-shot graph prompt learning setting as it has high performance on downstream tasks (as shown in Table 5 in Appendix A.4.1) and also high MIA risk (as shown in Figure 3).

**Privacy Parameters and Accounting.** We set the privacy parameters for `DP-GPL` according to Table 4 in Appendix A.2. Note that, since we scale the weights in `DP-GPL+W` to match the number of teachers (*i.e.,* the original sum of votes, see lines 6-8 in Algorithm 2), we can use the same parameters over both methods. To empirically account for the per-teacher privacy loss during our experiments, we build on the code-based from Boenisch et al. (2023).

**DP-GPL and DP-GPL+W.** We use an ensemble of 200 teacher prompts, and each teacher prompt is trained with disjoint 5 shots of data from the private downstream task. For query dataset, we select 50 public samples from the downstream distribution. Both methods are implemented to immediately stop querying once a teacher has reached their privacy limit, which we set to $\varepsilon = 2$. We repeat each experiment three times and report the average and standard deviation of the public student prompt's accuracy on the testing dataset.

**Baselines.** We compare against three baselines. (1) *Lower Bound (LB):* ($\varepsilon = 0$). Given a pre-trained GNN model, we directly evaluate its performance on the downstream test data. (2) *Ensemble Accuracy (Ens. Acc.):* ($\varepsilon = \infty$). We use the histogram of the private teacher ensemble votes and return the clean argmax. (3) *Upper Bound (UB):* ($\varepsilon = \infty$). *i.e.,* we select the teacher prompt which has the best testing accuracy.

### 5.2 RESULTS

We present the results of our `DP-GPL` and `DP-GPL+W`, and of the three baselines on different GNN models and downstream datasets in Table 1. The results for other graph prompt learning methods in Appendix A.4.4 show the same trends. We first observe that both our proposed algorithms significantly improve over the lower bound ($\varepsilon = 0$) baseline, highlighting their effectiveness in tuning graph prompts to solve the respective downstream tasks. While there is a slight performance gap, the test accuracies achieved by `DP-GPL+W` are generally close to the upper bound, *e.g.,* $64.64\%$ vs $67.12\%$ on Cora, GAT model. We furthermore observe that, in most cases, `DP-GPL+W` outperforms `DP-GPL`. Over the given setups, `DP-GPL+W` achieves, on average, a $3.5\%$ higher downstream utility than `DP-GPL`, while still consuming less than the specified privacy budget of $\varepsilon = 2$ over each teacher. This highlights that our data-aware partitioning algorithm and the weighting of the respective teachers according to the nodes' influence are effective in improving privacy-utility trade-offs.

Table 1: **Performance comparison between our `DP-GPL` & `DP-GPL+W`, and three baselines on three downstream datasets. (DGI, All-in-one, $\delta = 1.5 \times 10^{-4}$). LB – Lower Bound, UB – Upper Bound.** `DP-GPL` and `DP-GPL+W` perform significantly better than the lower bound in all setups, and `DP-GPL+W` has similar utility to the non-private baselines. For `DP-GPL+W`, we report the range of privacy consumptions experienced over all teachers. Generally, there is an improvement around 3.5% from `DP-GPL` to `DP-GPL+W`, indicating the effectiveness of weighted aggregation.

| | | LB | Ens. Acc. | UB | Our `DP-GPL` | | Our `DP-GPL+W` | |
|---|---|---|---|---|---|---|---|---|
| | Private | $\varepsilon = 0$ | $\varepsilon = \infty$ | $\varepsilon = \infty$ | $\varepsilon$ | Test Acc | $\varepsilon_{max}$ | Test Acc |
| GAT | Cora | 43.92 | 67.09 | 67.12 | 0.2226 | 57.96 $\pm 2.12$ | 1.6247 | 64.64 $\pm 0.80$ |
| | CiteSeer | 37.51 | 73.44 | 74.75 | 0.2047 | 73.49 $\pm 2.04$ | 1.6078 | 71.45 $\pm 2.06$ |
| | PubMed | 32.86 | 71.48 | 71.72 | 0.2383 | 66.07 $\pm 1.78$ | 1.6555 | 68.17 $\pm 6.15$ |
| GCN | Cora | 49.10 | 62.35 | 64.04 | 0.2025 | 56.22 $\pm 2.00$ | 1.6859 | 61.30 $\pm 1.38$ |
| | CiteSeer | 40.51 | 62.95 | 64.63 | 0.2001 | 59.41 $\pm 1.97$ | 1.6244 | 61.76 $\pm 2.06$ |
| | PubMed | 29.95 | 69.09 | 70.13 | 0.2386 | 62.70 $\pm 2.10$ | 1.6276 | 67.94 $\pm 3.02$ |
| GT | Cora | 21.80 | 55.36 | 56.77 | 0.2276 | 54.53 $\pm 1.97$ | 1.7053 | 53.91 $\pm 0.47$ |
| | CiteSeer | 27.56 | 51.75 | 53.51 | 0.3627 | 43.88 $\pm 2.13$ | 1.7392 | 50.04 $\pm 2.70$ |
| | PubMed | 39.23 | 70.63 | 72.95 | 0.2084 | 63.93 $\pm 2.15$ | 1.5999 | 70.26 $\pm 3.00$ |

Regarding our methods' privacy consumption, we observe that neither exhausts the given privacy budget of $\varepsilon = 2$. In particular, `DP-GPL` is not able to spend above $\varepsilon = 0.3627$ during the labeling. This small privacy consumption is due to the limited number of public samples used for the knowledge transfer: over the given 50 queries, the methods cannot spend more privacy. While it would be possible to increase the number of public queries, we find that this does not increase the downstream performance notably. Hence, by limiting the public data to 50 samples, the best privacy-utility trade-offs can be achieved. As for `DP-GPL+W`, we see that it is able to consume the given privacy budget more effectively over the 50 queries, by spending up to $\varepsilon = 1.7392$ in the teachers with the highest weight—yielding a significant improvement in downstream performance. This shows that, by enabling `DP-GPL+W` to spend privacy non-uniformly over the teachers, it can benefit from the given privacy budget where `DP-GPL` is not able to spend it—causing lower utility.

## 6 CONCLUSIONS

In this work, we are the first to highlight the privacy risks that arise from graph prompt learning. By running a membership inference attack, we showed that private information from the private dataset used to tune the graph prompts can leak to external parties who query the prompted GNN. To mitigate the resulting risk for the downstream data, we set out to design a private graph prompt learning algorithm. Motivated by our finding that the naive application of the DP-SGD algorithm, the standard to implement DP guarantees in machine learning, fails to yield good privacy-utility trade-offs, we designed `DP-GPL` and `DP-GPL+W`, which build on the PATE algorithm and perform a noisy knowledge transfer from teachers to a student prompt. Leveraging the natural structure of graph data, in contrast to standard PATE, in `DP-GPL+W`, we do not weight every teacher's vote equally during the knowledge transfer. Instead, we weight teachers who hold nodes with higher centrality (more influential nodes) higher. We thoroughly analyzed the resulting utility and privacy implications and highlighted that our `DP-GPL` and `DP-GPL+W` are able to yield strong utility at high privacy guarantees. Thereby, our work contributes towards leveraging the computational and utility benefits from graph prompt learning but without additional privacy risks for the downstream data.

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

## A APPENDIX

### A.1 EXPERIMENTAL SETUP: DATASETS

Table 2: **Statistics of datasets.** $|\mathcal{V}|, |\mathcal{E}|, m, |\mathbb{C}|$ denote the number of nodes, num of edges, dimension of a node feature vector, and number of classes, respectively.

| Dataset | $|\mathcal{V}|$ | $|\mathcal{E}|$ | $m$ | $|\mathbb{C}|$ |
|---|---|---|---|---|
| ogbn-arxiv | $169,343$ | $1,166,243$ | $128$ | $40$ |
| Cora | $2,708$ | $10,556$ | $1,433$ | $7$ |
| CiteSeer | $3,327$ | $9,104$ | $3,703$ | $6$ |
| PubMed | $19,717$ | $88,648$ | $500$ | $3$ |

In this paper, we focus on graph prompt learning for node-level tasks. Also, we consider the scenario where a GNN model is pretrained on a large graph by the model provider, and users apply it to a specific downstream task (a smaller graph) through graph prompt learning (Sun et al., 2023b). To simulate this scenario, we use ogbn-arxiv, which is a large-scale graph dataset, as the pre-training dataset. For the downstream tasks, we use Cora, CiteSeer, and PubMed (Yang et al., 2016). The statistics of datasets are presented in Table 2.

## A.2 EXPERIMENTAL SETUP: HYPERPARAMETERS

The default hyperparameters used in the GNN pre-training phase are presented in Table 3. And Table 4 shows the parameters for Confident-GNMax used in `DP-GPL` and `DP-GPL+W`.

Table 3: **Default hyperparameter setting for GNN pre-training.**

| Type | Hyperparameter | Setting |
|---|---|---|
| GAT | Architecture | 3 layers |
| | Hidden unit size | 128 |
| GCN | Architecture | 3 layers |
| | Hidden unit size | 128 |
| Graph Transformer | Architecture | 3 layers |
| | Hidden unit size | 128 |
| Training | Learning rate | 0.001 |
| | Optimizer | Adam |
| | Epochs | 300 |
| | Batch size | 128 |

Table 4: **Parameters for Confident-GNMax.** ($T$ - threshold, $\sigma_1$, $\sigma_2$ - noise parameters)

| GNN model | Downsteream dataset | $T$ | $\sigma_1$ | $\sigma_2$ |
|---|---|---|---|---|
| GAT | Cora | 170 | 5 | 100 |
| GAT | CiteSeer | 170 | 5 | 50 |
| GAT | PubMed | 170 | 1 | 20 |
| GCN | Cora | 150 | 1 | 20 |
| GCN | CiteSeer | 180 | 1 | 20 |
| GCN | PubMed | 170 | 1 | 20 |
| GT | Cora | 150 | 10 | 100 |
| GT | CiteSeer | 150 | 5 | 50 |
| GT | PubMed | 170 | 5 | 100 |

## A.3 PSEUDOCODE FOR OUR `DP-GPL`

We here provide the pseudocode for our `DP-GPL` and `DP-GPL+W` algorithm in Algorithm 3. This algorithm includes the main five steps in our methods, *i.e.,* private data partition, teacher prompts training, prompting pre-trained GNN model, aggregation, and student prompt training. In this algorithm, we highlight the difference between our methods and the standard PATE in blue.

## A.4 ADDITIONAL EXPERIMENTS

### A.4.1 PERFORMANCE OF GRAPH PROMPT LEARNING

One advantage of graph prompt learning is that in the few-shot setting, the downstream performance of graph prompt learning is comparable to or even better than the "pre-train & fine-tune" paradigm. We implement preliminary experiments to compare the downstream performance of graph prompt learning and the fine-tuning paradigm in a 5-shot setting, as shown in Table 5. As we can see, in most cases, the testing accuracy of graph prompt methods is close to or higher than that of the fine-tuning paradigm, making it reasonable to explore the privacy risk of graph prompt learning in the few-shot setting.

### A.4.2 MIA RESULTS

Figure 5 and Figure 6 show our MIA on CiteSeer and PubMed datasets, respectively, with 1-5 shots of private data used in training prompts. As we can observe, our MIA has higher attack success with few shots.

**Algorithm 3 `DP-GPL` & `DP-GPL+W`.** In contrast to the standard PATE algorithm where the teacher models are trained on disjoint subsets of private data, our `DP-GPL` and `DP-GPL+W` trains teacher prompts on disjoint subsets of the private graph data. Also, it's notable that in our `DP-GPL+W`, we partition private data based on centrality scores and utilize weighted aggregation mechanism. We highlight these differences in blue.

**Require:** Private graph data $V_{private} = \{(x_1, y_1), (x_2, y_2), \ldots, (x_n, y_n)\}$
**Require:** Number of teachers $N$, threshold $T$, noise parameters $\sigma_1$ and $\sigma_2$, maximum weight $w_{\max}$ and minimum weight $w_{\min}$
**Require:** Pre-trained GNN model $\Phi$, unlabeled public query data $V_{public}$
1: **Step 1: Private data partition**
2: /* `DP-GPL` */
3: Partition $V_{private}$ into $N$ IID disjoint groups $\{g_1, g_2, \ldots, g_N\}$
4: /* `DP-GPL+W` */
5: Calculate *centrality* score of each node in $V_{private}$
6: Partition $V_{private}$ into $N$ disjoint groups $\{g_1, g_2, \ldots, g_N\}$ according to the centrality scores
7: Get average centrality score of each subset: $\mathcal{S} = \{s_1, s_2, \ldots, s_N\}$
8: **for** each teacher $i = 1$ to $N$ **do**
9:     **Step 2: Teacher Prompts Training**
10:     Train teacher prompt $\mathcal{P}_i$ on the group $g_i$
11: **end for**
12: **Step 3: Prompting pre-trained GNN model**
13: Actual public data $D_{public} = \varnothing$
14: **for** each query $x_j \in V_{public}$ (e.g., a node) **do**
15:     Insert teacher prompt $\mathcal{P}_i$ into the query data point, *i.e.,* $\mathcal{P}_i(x_j)$
16:     Query the pre-trained GNN model and get a label $y_i^j = \Phi(\mathcal{P}_i(x_j))$
17:     **Step 4: Aggregation**
18:     /* `DP-GPL` */
19:     Get count for each class with uniform votes: $c_m(x_j) = \sum_i^N (y_i^j = m)$
20:     /* `DP-GPL+W` */
21:     Get count for each class with weighted votes: $c_m(x_j) = \sum_i^N w_i \cdot (y_i^j = m)$
22:     **if** $max_m \{c_m(x_j)\} + \mathcal{N}(0, \sigma_1^2) \geq T$ **then**            $\triangleright$ $m$ is the class label
23:         $y^j = \arg\max_m \{c_m(x_j) + \mathcal{N}(0, \sigma_2^2)\}$
24:         $D_{public} = D_{public} \cup (x_j, y_j)$
25:     **end if**
26: **end for**
27: **Step 5: Student Prompt Training**
28: Train student prompt $\mathcal{P}_s$ using the noisy labeled public data $D_{public}$
29: **Differential Privacy Guarantee**
30: Compute actual privacy loss $(\varepsilon, \delta)$ based on noise parameters $\sigma_1$, $\sigma_2$ and the number of queries $|D_{public}|$
31: **return** Student prompt $\mathcal{P}_s$ with differentially private guarantee

Table 5: **Performance of Pre-train & Fine-tune (PFT) and graph prompt learning (Cora, 5-shot).**

| GNN architectures | Pre-train methods | PFT | All-in-one | GPF-plus | GPPT |
|---|---|---|---|---|---|
| GAT | DGI | 46.03 $\pm 0.79$ | 48.70 $\pm 1.45$ | 53.48 $\pm 1.99$ | 56.53 $\pm 1.51$ |
| | EdgePreGPPT | 56.33 $\pm 1.29$ | 48.71 $\pm 1.11$ | 40.89 $\pm 1.53$ | 54.77 $\pm 1.54$ |
| | GraphMAE | 43.51 $\pm 0.74$ | 50.66 $\pm 1.03$ | 51.61 $\pm 1.09$ | 49.32 $\pm 1.49$ |
| | SimGRACE | 14.71 $\pm 1.67$ | 13.05 $\pm 1.62$ | 21.35 $\pm 1.24$ | 35.03 $\pm 2.07$ |
| GCN | DGI | 52.12 $\pm 1.36$ | 58.25 $\pm 1.10$ | 66.50 $\pm 2.50$ | 56.21 $\pm 1.68$ |
| | EdgePreGPPT | 43.77 $\pm 1.16$ | 68.94 $\pm 1.09$ | 76.30 $\pm 0.98$ | 60.28 $\pm 1.86$ |
| | GraphMAE | 39.55 $\pm 1.24$ | 62.90 $\pm 0.91$ | 75.84 $\pm 1.10$ | 51.63 $\pm 1.25$ |
| | SimGRACE | 18.15 $\pm 0.52$ | 18.19 $\pm 1.64$ | 19.97 $\pm 0.65$ | 33.72 $\pm 1.98$ |
| GraphTransformer | DGI | 53.33 $\pm 1.09$ | 45.12 $\pm 2.05$ | 29.54 $\pm 2.24$ | 56.21 $\pm 1.51$ |
| | EdgePreGPPT | 60.02 $\pm 1.07$ | 53.45 $\pm 1.06$ | 35.74 $\pm 0.59$ | 56.95 $\pm 1.04$ |
| | GraphMAE | 52.95 $\pm 1.44$ | 41.84 $\pm 0.97$ | 36.58 $\pm 0.67$ | 48.54 $\pm 1.17$ |
| | SimGRACE | 39.79 $\pm 0.25$ | 15.03 $\pm 1.12$ | 15.60 $\pm 0.88$ | 41.14 $\pm 0.57$ |

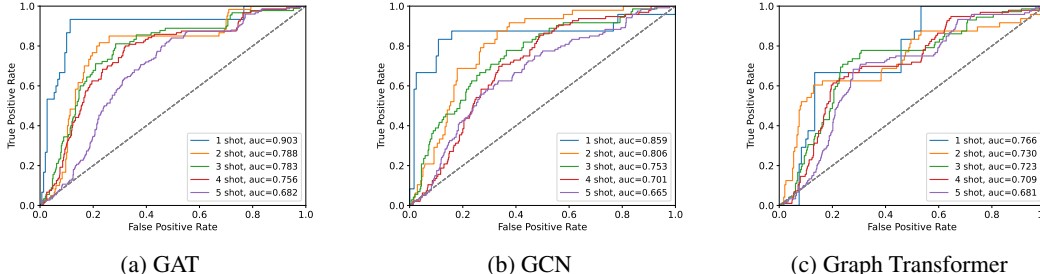

|  (a) GAT | (b) GCN | (c) Graph Transformer |

Figure 5: **AUC-ROC curve of our MIA on CiteSeer dataset with different number of shots,** *i.e.,* **1-5 shots.**

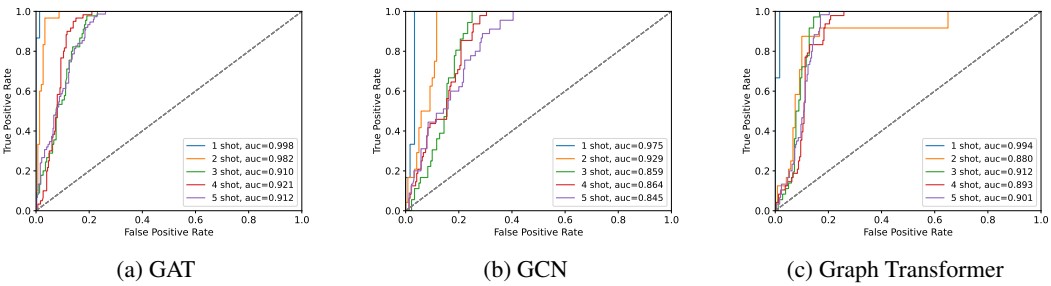

|  (a) GAT | (b) GCN | (c) Graph Transformer |

Figure 6: **AUC-ROC curve of our MIA attack on PubMed dataset with different number of shots,** *i.e.,* **1-5 shots.**

### A.4.3 RESULTS OF DP-SGD ON GRAPH PROMPT LEARNING

Table 6 shows the performance of DP-SGD on graph prompt learning with different privacy budgets and numbers of shots. It is evident that the DP-SGD algorithm significantly degrades the downstream task's performance even at high privacy budgets. Only when the number of shots increases to 100, the DP-SGD algorithm can achieve a high utility. However, in the few-shot setting (*i.e.,* less than 50 shots), the DP-SGD algorithm fails to have a great privacy-utility trade-off.

Table 6: **Performance of DP-SGD on graph prompt learning on Cora dataset (DGI, GPF-plus, GAT).**

| # Shots | $\varepsilon = \infty$ | $\varepsilon = 1$ | $\varepsilon = 8$ | $\varepsilon = 16$ | $\varepsilon = 32$ | $\varepsilon = 64$ |
|---|---|---|---|---|---|---|
| 5 | 48.70 $\pm 1.45$ | 15.10 $\pm 1.09$ | 15.46 $\pm 1.13$ | 16.58 $\pm 0.17$ | 17.04 $\pm 1.01$ | 18.47 $\pm 0.91$ |
| 10 | 65.70 $\pm 5.15$ | 17.04 $\pm 0.43$ | 16.75 $\pm 3.29$ | 17.33 $\pm 2.91$ | 18.09 $\pm 0.21$ | 18.67 $\pm 0.96$ |
| 50 | 75.20 $\pm 2.09$ | 19.58 $\pm 0.17$ | 19.91 $\pm 3.15$ | 22.55 $\pm 1.63$ | 22.44 $\pm 2.04$ | 22.04 $\pm 1.20$ |
| 100 | 78.42 $\pm 0.98$ | 68.15 $\pm 0.94$ | 77.27 $\pm 0.33$ | 77.94 $\pm 1.85$ | 78.16 $\pm 1.94$ | 78.40 $\pm 1.53$ |

A.4.4 `DP-GPL` & `DP-GPL+W` RESULTS

We also present the performance of our `DP-GPL` and `DP-GPL+W` on other setups, see Table 7 to Table 11. In addition, we present the full privacy cost range of `DP-GPL+W` in Table 12, as an addition to Table 1. In consistent with the observations in Section 5.2, our `DP-GPL` and `DP-GPL+W` can achieve high utility under strong privacy guarantees. And `DP-GPL+W` can achieve better utility than `DP-GPL` in most cases, indicating the effectiveness of our weighted aggregation mechanism.

Table 7: **Performance comparison between our `DP-GPL` & `DP-GPL+W` and three baselines on three downstream datasets. (DGI, GPF-plus, $\delta = 1.5 \times 10^{-4}$).** LB – Lower Bound, UB – Upper Bound.

| | | LB | Ens. Acc. | UB | our `DP-GPL` | | our `DP-GPL+W` | |
|---|---|---|---|---|---|---|---|---|
| | Private | $\varepsilon = 0$ | $\varepsilon = \infty$ | $\varepsilon = \infty$ | $\varepsilon$ | Test Acc | $\varepsilon_{max}$ | Test Acc |
| GAT | Cora | 43.92 | 59.14 | 60.13 | 0.9186 | 58.10 $\pm 1.63$ | 1.6884 | 59.00 $\pm 0.88$ |
| | CiteSeer | 37.51 | 69.24 | 70.38 | 0.4917 | 68.11 $\pm 1.39$ | 1.6124 | 69.83 $\pm 0.69$ |
| | PubMed | 32.86 | 79.07 | 79.22 | 0.3150 | 78.85 $\pm 1.40$ | 1.5789 | 78.41 $\pm 0.91$ |
| GCN | Cora | 49.10 | 71.33 | 77.87 | 0.4268 | 64.64 $\pm 0.73$ | 1.7456 | 77.14 $\pm 3.34$ |
| | CiteSeer | 40.51 | 82.70 | 85.98 | 0.2039 | 79.44 $\pm 5.74$ | 1.7968 | 84.55 $\pm 1.72$ |
| | PubMed | 29.95 | 80.76 | 81.73 | 0.2486 | 79.81 $\pm 5.17$ | 1.6712 | 80.94 $\pm 0.99$ |
| GT | Cora | 21.80 | 37.81 | 38.08 | 0.9990 | 37.38 $\pm 1.69$ | 1.6852 | 36.83 $\pm 0.25$ |
| | CiteSeer | 27.56 | 37.78 | 37.88 | 0.9933 | 37.61 $\pm 3.04$ | 1.6414 | 35.87 $\pm 3.87$ |
| | PubMed | 39.23 | 71.17 | 73.45 | 0.9973 | 68.94 $\pm 0.94$ | 1.6971 | 71.82 $\pm 1.24$ |

Table 8: **Performance comparison between our `DP-GPL` & `DP-GPL+W`, and three baselines on three downstream datasets. (DGI, GPPT, $\delta = 1.5 \times 10^{-4}$).** LB – Lower Bound, UB – Upper Bound.

| | | LB | Ens. Acc. | UB | our `DP-GPL` | | our `DP-GPL+W` | |
|---|---|---|---|---|---|---|---|---|
| | Private | $\varepsilon = 0$ | $\varepsilon = \infty$ | $\varepsilon = \infty$ | $\varepsilon$ | Test Acc | $\varepsilon_{max}$ | Test Acc |
| GAT | Cora | 43.92 | 51.73 | 56.39 | 0.7777 | 46.90 $\pm 1.24$ | 1.5933 | 52.68 $\pm 0.86$ |
| | CiteSeer | 37.51 | 48.55 | 54.29 | 0.4790 | 42.65 $\pm 1.26$ | 1.6450 | 51.83 $\pm 8.87$ |
| | PubMed | 32.86 | 63.97 | 68.25 | 0.2874 | 59.55 $\pm 0.88$ | 1.6472 | 65.56 $\pm 3.28$ |
| GCN | Cora | 49.10 | 59.23 | 64.16 | 0.4980 | 54.15 $\pm 2.02$ | 1.5631 | 61.54 $\pm 3.51$ |
| | CiteSeer | 40.51 | 56.41 | 60.60 | 0.3728 | 52.09 $\pm 1.19$ | 1.6829 | 57.43 $\pm 4.31$ |
| | PubMed | 29.95 | 68.41 | 73.41 | 0.2601 | 63.28 $\pm 4.75$ | 1.6774 | 69.63 $\pm 0.54$ |
| GT | Cora | 21.80 | 56.84 | 58.74 | 0.6964 | 54.78 $\pm 3.15$ | 1.7367 | 57.22 $\pm 0.17$ |
| | CiteSeer | 27.56 | 48.28 | 49.76 | 0.5904 | 46.63 $\pm 2.86$ | 1.7246 | 46.08 $\pm 0.56$ |
| | PubMed | 39.23 | 66.52 | 69.46 | 0.3846 | 63.38 $\pm 2.11$ | 1.5895 | 66.14 $\pm 1.83$ |

Table 9: **Performance comparison between our `DP-GPL` & `DP-GPL+W` and three baselines on three downstream datasets. (GraphMAE, All-in-one, $\delta = 1.5 \times 10^{-4}$).** LB – Lower Bound, UB – Upper Bound.

| | | LB | Ens. Acc. | UB | our `DP-GPL` | | our `DP-GPL+W` | |
|---|---|---|---|---|---|---|---|---|
| | Private | $\varepsilon = 0$ | $\varepsilon = \infty$ | $\varepsilon = \infty$ | $\varepsilon$ | Test Acc | $\varepsilon_{max}$ | Test Acc |
| GAT | Cora | 39.65 | 49.40 | 52.94 | 0.5728 | 41.02 $\pm 1.38$ | 1.6222 | 47.76 $\pm 2.09$ |
| | CiteSeer | 38.50 | 39.09 | 40.87 | 0.2412 | 29.27 $\pm 2.10$ | 1.7290 | 36.71 $\pm 1.58$ |
| | PubMed | 30.86 | 64.64 | 67.85 | 0.2232 | 58.81 $\pm 0.59$ | 1.6265 | 62.66 $\pm 2.52$ |
| GCN | Cora | 30.76 | 62.97 | 65.37 | 0.0782 | 59.50 $\pm 0.63$ | 1.6897 | 60.75 $\pm 1.13$ |
| | CiteSeer | 31.85 | 67.89 | 71.85 | 0.0588 | 61.68 $\pm 0.41$ | 1.8023 | 65.22 $\pm 0.49$ |
| | PubMed | 32.87 | 70.22 | 71.46 | 0.4989 | 64.59 $\pm 0.11$ | 1.8862 | 67.95 $\pm 1.83$ |
| GT | Cora | 35.68 | 47.35 | 48.65 | 0.4197 | 37.47 $\pm 1.05$ | 1.5340 | 44.68 $\pm 0.63$ |
| | CiteSeer | 34.67 | 52.58 | 56.48 | 0.0390 | 46.97 $\pm 2.18$ | 1.8430 | 50.15 $\pm 0.25$ |
| | PubMed | 22.38 | 34.34 | 35.47 | 0.3359 | 32.82 $\pm 1.42$ | 1.7741 | 31.60 $\pm 4.30$ |

A.4.5 INFLUENCE OF THE NUMBER OF QUERIES

We analyze the impact of the number of public queries on the performance of our `DP-GPL` and `DP-GPL+W` in Figure 7, taking Cora, DGI, All-in-one, and GAT as an example. As we can see, the

Table 10: **Performance comparison between our DP-GPL & DP-GPL+W and three baselines on three downstream datasets. (GraphMAE, GPF-plus, $\delta = 1.5 \times 10^{-4}$).** LB – Lower Bound, UB – Upper Bound.

| | | LB | Ens. Acc. | UB | our DP-GPL | | our DP-GPL+W | |
|---|---|---|---|---|---|---|---|---|
| | Private | $\varepsilon = 0$ | $\varepsilon = \infty$ | $\varepsilon = \infty$ | $\varepsilon$ | Test Acc | $\varepsilon_{max}$ | Test Acc |
| GAT | Cora | 39.65 | 51.69 | 54.38 | 0.6778 | 45.44 ±7.09 | 1.9746 | 49.79 ±0.95 |
| | CiteSeer | 38.50 | 58.02 | 61.94 | 0.2194 | 54.50 ±3.41 | 1.9904 | 56.80 ±2.63 |
| | PubMed | 30.86 | 76.21 | 78.56 | 0.4846 | 66.21 ±3.05 | 1.8690 | 75.16 ±1.77 |
| GCN | Cora | 30.76 | 74.16 | 76.85 | 0.6135 | 66.88 ±1.91 | 1.9611 | 74.09 ±1.76 |
| | CiteSeer | 31.85 | 78.13 | 80.87 | 0.6262 | 69.45 ±2.58 | 1.9383 | 77.27 ±3.42 |
| | PubMed | 32.87 | 77.84 | 80.85 | 0.0595 | 68.67 ±5.32 | 1.7512 | 77.02 ±0.10 |
| GT | Cora | 35.68 | 39.10 | 42.49 | 0.0273 | 30.78 ±3.33 | 1.6369 | 38.15 ±0.81 |
| | CiteSeer | 34.67 | 41.61 | 43.75 | 0.1189 | 34.72 ±0.54 | 1.8614 | 40.92 ±1.09 |
| | PubMed | 22.38 | 28.29 | 31.39 | 0.6147 | 21.86 ±1.69 | 1.8231 | 27.31 ±3.09 |

Table 11: **Performance comparison between our DP-GPL & DP-GPL+W and three baselines on three downstream datasets. (GraphMAE, GPPT, $\delta = 1.5 \times 10^{-4}$).** LB – Lower Bound, UB – Upper Bound.

| | | LB | Ens. Acc. | UB | our DP-GPL | | our DP-GPL+W | |
|---|---|---|---|---|---|---|---|---|
| | Private | $\varepsilon = 0$ | $\varepsilon = \infty$ | $\varepsilon = \infty$ | $\varepsilon$ | Test Acc | $\varepsilon_{max}$ | Test Acc |
| GAT | Cora | 39.65 | 49.99 | 51.57 | 0.5979 | 47.08 ±2.13 | 1.8589 | 47.65 ±3.54 |
| | CiteSeer | 38.50 | 45.44 | 46.48 | 0.6392 | 43.02 ±0.41 | 1.5596 | 43.43 ±0.69 |
| | PubMed | 30.86 | 55.48 | 56.64 | 0.5325 | 53.92 ±0.05 | 1.7982 | 53.87 ±1.34 |
| GCN | Cora | 30.76 | 54.57 | 54.85 | 0.3617 | 51.61 ±3.98 | 1.5649 | 52.09 ±0.64 |
| | CiteSeer | 31.85 | 44.12 | 45.78 | 0.1175 | 41.53 ±1.17 | 1.5388 | 41.77 ±0.12 |
| | PubMed | 32.87 | 59.25 | 60.57 | 0.2091 | 56.35 ±1.64 | 1.9156 | 57.77 ±2.09 |
| GT | Cora | 35.68 | 52.63 | 54.09 | 0.1988 | 50.24 ±4.11 | 1.7322 | 50.81 ±2.02 |
| | CiteSeer | 34.67 | 65.16 | 65.78 | 0.2290 | 63.49 ±5.21 | 1.5810 | 63.73 ±0.88 |
| | PubMed | 22.38 | 46.41 | 47.97 | 0.3221 | 44.03 ±3.00 | 1.7740 | 45.07 ±2.26 |

performance of our DP-GPL and DP-GPL+W increases as the number of public queries increases from 10 to 50. With more than 50 public queries, the performance of our DP-GPL and DP-GPL+W tends to be stable, indicating that our methods can achieve the best privacy-utility trade-offs with 50 public queries.

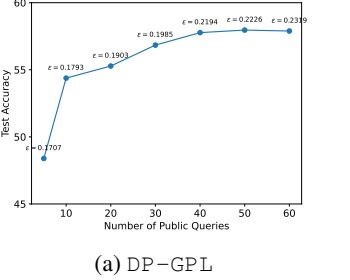
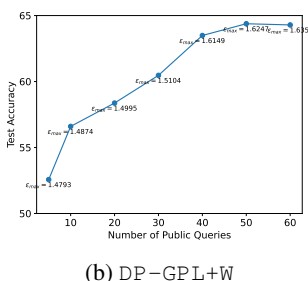

(a) DP-GPL  (b) DP-GPL+W

Figure 7: **Influence of the number of public queries on the performance of our DP-GPL and DP-GPL+W (Cora, DGI, All-in-one, GAT).**

### A.4.6 INFLUENCE OF THE MAXIMUM WEIGHT

We analyze the impact of the maximum weight - $w_{max}$ in our weighted aggregation mechanism on the performance of our DP-GPL+W. The results of DP-GPL+W with $w_{max} = 1.5, 2.5, 3.0$ with DGI, All-in-one are shown in Table 13, Table 14 and Table 15 respectively. The comparison of the performance of our DP-GPL+W with different $w_{max}$ is shown in Figure 8. We can observe that with $w_{max} = 2.0$, DP-GPL+W can achieve the best privacy-utility trade-offs.

Table 12: **Performance comparison between our `DP-GPL` & `DP-GPL+W`, and three baselines on three downstream datasets, with $\varepsilon$ range for `DP-GPL+W`. (DGI, All-in-one, $\delta = 1.5 \times 10^{-4}$). LB – Lower Bound, UB – Upper Bound.**

| | | LB | Ens. Acc. | UB | Our `DP-GPL` | | Our `DP-GPL+W` | |
|---|---|---|---|---|---|---|---|---|
| | Private | $\varepsilon = 0$ | $\varepsilon = \infty$ | $\varepsilon = \infty$ | $\varepsilon$ | Test Acc | $\varepsilon_i$ | Test Acc |
| GAT | Cora | 43.92 | 67.09 | 67.12 | 0.2226 | 57.96 $\pm 2.12$ | [0.1015, 1.6247] | 64.64 $\pm 0.80$ |
| | CiteSeer | 37.51 | 73.44 | 74.75 | 0.2047 | 73.49 $\pm 2.04$ | [0.1005, 1.6078] | 71.45 $\pm 2.06$ |
| | PubMed | 32.86 | 71.48 | 71.72 | 0.2383 | 66.07 $\pm 1.78$ | [0.1035, 1.6555] | 68.17 $\pm 6.15$ |
| GCN | Cora | 49.10 | 62.35 | 64.04 | 0.2025 | 56.22 $\pm 2.00$ | [0.1054, 1.6859] | 61.30 $\pm 1.38$ |
| | CiteSeer | 40.51 | 62.95 | 64.63 | 0.2001 | 59.41 $\pm 1.97$ | [0.1015, 1.6244] | 61.76 $\pm 2.06$ |
| | PubMed | 29.95 | 69.09 | 70.13 | 0.2386 | 62.70 $\pm 2.10$ | [0.1017, 1.6276] | 67.94 $\pm 3.02$ |
| GT | Cora | 21.80 | 55.36 | 56.77 | 0.2276 | 54.53 $\pm 1.97$ | [0.1066, 1.7053] | 53.91 $\pm 0.47$ |
| | CiteSeer | 27.56 | 51.75 | 53.51 | 0.3627 | 43.88 $\pm 2.13$ | [0.1087, 1.7392] | 50.04 $\pm 2.70$ |
| | PubMed | 39.23 | 70.63 | 72.95 | 0.2084 | 63.93 $\pm 2.15$ | [0.1000, 1.5999] | 70.26 $\pm 3.00$ |

Table 13: **Performance comparison between `DP-GPL` and three baselines on three downstream datasets. (DGI, All-in-one, $w_{max} = 1.5$). LB – Lower Bound, UB – Upper Bound.**

| | | LB | Ens. Acc. | UB | our `DP-GPL` | | our `DP-GPL+W` | |
|---|---|---|---|---|---|---|---|---|
| | Private | $\varepsilon = 0$ | $\varepsilon = \infty$ | $\varepsilon = \infty$ | $\varepsilon$ | Test Acc | $\varepsilon_i$ | Test Acc |
| GAT | Cora | 43.92 | 67.09 | 67.12 | 0.2226 | 57.96 $\pm 2.12$ | [0.1600, 1.4400] | 58.84 $\pm 0.99$ |
| | CiteSeer | 37.51 | 73.44 | 74.75 | 0.2047 | 73.49 $\pm 2.04$ | [0.1544, 1.3899] | 72.83 $\pm 3.58$ |
| | PubMed | 55.82 | 71.48 | 71.72 | 0.2383 | 66.07 $\pm 1.78$ | [0.1547, 1.3927] | 70.93 $\pm 0.61$ |
| GCN | Cora | 49.10 | 62.35 | 64.04 | 0.2025 | 56.22 $\pm 2.00$ | [0.1542, 1.3880] | 55.72 $\pm 1.08$ |
| | CiteSeer | 40.51 | 62.95 | 64.63 | 0.2001 | 59.41 $\pm 1.97$ | [0.1555, 1.3991] | 59.63 $\pm 1.81$ |
| | PubMed | 57.84 | 69.09 | 70.13 | 0.2386 | 62.70 $\pm 2.10$ | [0.1524, 1.3716] | 65.86 $\pm 4.99$ |
| GT | Cora | 21.80 | 55.36 | 56.77 | 0.2276 | 54.53 $\pm 1.97$ | [0.1528, 1.3748] | 52.71 $\pm 2.10$ |
| | CiteSeer | 27.56 | 51.75 | 53.51 | 0.3627 | 43.88 $\pm 2.13$ | [0.1661, 1.4946] | 42.70 $\pm 3.45$ |
| | PubMed | 39.23 | 70.63 | 72.95 | 0.2084 | 63.93 $\pm 2.15$ | [0.1660, 1.4940] | 64.31 $\pm 5.57$ |

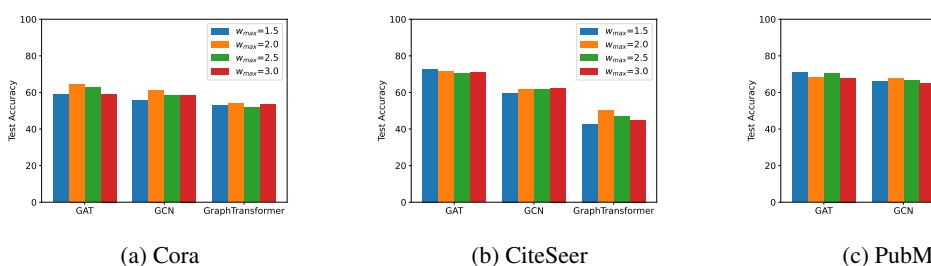

(a) Cora        (b) CiteSeer        (c) PubMed

Figure 8: **Influence of the maximum weight on the performance of our `DP-GPL+W` (DGI, All-in-one).**

### A.4.7 DISTRIBUTION OF $\varepsilon$ OVER TEACHER ENSEMBLES

We provide the distribution of $\varepsilon$ over teacher ensembles in Figure 9, taking DGI, All-in-one, and GAT as an example. As we can see, the distribution of $\varepsilon$ is mostly concentrated around a small value, with a small number of teachers having a large $\varepsilon$. It indicates that `DP-GPL+W` can consume the given privacy budget effectively and achieve high utility.

### A.4.8 MIA RESULTS AGAINST `DP-GPL` AND `DP-GPL+W`

We also evaluate the effectiveness of our `DP-GPL` and `DP-GPL+W` against MIA, as shown in Figure 10 and Figure 11. The member data is the private data used in training all teacher prompts, and the non-members are randomly selected samples from the testing dataset. As we can see, all curves are very close to the dash line (random guess), which shows that our `DP-GPL` and `DP-GPL+W` are effective against MIA, for all downstream tasks and GNN architectures.

Table 14: **Performance comparison between DP-GPL and three baselines on three downstream datasets. (DGI, All-in-one, $w_{max} = 2.5$).** LB – Lower Bound, UB – Upper Bound.

|  |  | LB | Ens. Acc. | UB | our DP-GPL | | our DP-GPL+W | |
|---|---|---|---|---|---|---|---|---|
|  | Private | $\varepsilon = 0$ | $\varepsilon = \infty$ | $\varepsilon = \infty$ | $\varepsilon$ | Test Acc | $\varepsilon_i$ | Test Acc |
| GAT | Cora | 43.92 | 67.09 | 67.12 | 0.2226 | 57.96 $\pm 2.12$ | [0.0974, 2.4341] | 62.64 $\pm 3.07$ |
|  | CiteSeer | 37.51 | 73.44 | 74.75 | 0.2047 | 73.49 $\pm 2.04$ | [0.0971, 2.4273] | 70.44 $\pm 6.09$ |
|  | PubMed | 55.82 | 71.48 | 71.72 | 0.2383 | 66.07 $\pm 1.78$ | [0.0954, 2.3852] | 70.53 $\pm 0.40$ |
| GCN | Cora | 49.10 | 62.35 | 64.04 | 0.2025 | 56.22 $\pm 2.00$ | [0.0912, 2.2812] | 58.35 $\pm 5.18$ |
|  | CiteSeer | 40.51 | 62.95 | 64.63 | 0.2001 | 59.41 $\pm 1.97$ | [0.0996, 2.4894] | 61.73 $\pm 5.55$ |
|  | PubMed | 57.84 | 69.09 | 70.13 | 0.2386 | 62.70 $\pm 2.10$ | [0.0940, 2.3508] | 66.49 $\pm 3.01$ |
| GT | Cora | 21.80 | 55.36 | 56.77 | 0.2276 | 54.53 $\pm 1.97$ | [0.0922, 2.3042] | 51.77 $\pm 2.97$ |
|  | CiteSeer | 27.56 | 51.75 | 53.51 | 0.3627 | 43.88 $\pm 2.13$ | [0.0972, 2.4293] | 47.14 $\pm 0.69$ |
|  | PubMed | 39.23 | 70.63 | 72.95 | 0.2084 | 63.93 $\pm 2.15$ | [0.0999, 2.4986] | 66.94 $\pm 2.63$ |

Table 15: **Performance comparison between DP-GPL and three baselines on three downstream datasets. (DGI, All-in-one, $w_{max} = 3.0$).** LB – Lower Bound, UB – Upper Bound.

|  |  | LB | Ens. Acc. | UB | our DP-GPL | | our DP-GPL+W | |
|---|---|---|---|---|---|---|---|---|
|  | Private | $\varepsilon = 0$ | $\varepsilon = \infty$ | $\varepsilon = \infty$ | $\varepsilon$ | Test Acc | $\varepsilon_i$ | Test Acc |
| GAT | Cora | 43.92 | 67.09 | 67.12 | 0.2226 | 57.96 $\pm 2.12$ | [0.0752, 2.7075] | 59.11 $\pm 6.08$ |
|  | CiteSeer | 37.51 | 73.44 | 74.75 | 0.2047 | 73.49 $\pm 2.04$ | [0.0800, 2.8802] | 70.88 $\pm 2.49$ |
|  | PubMed | 55.82 | 71.48 | 71.72 | 0.2383 | 66.07 $\pm 1.78$ | [0.0829, 2.9850] | 67.95 $\pm 1.43$ |
| GCN | Cora | 49.10 | 62.35 | 64.04 | 0.2025 | 56.22 $\pm 2.00$ | [0.0769, 2.7691] | 58.67 $\pm 3.65$ |
|  | CiteSeer | 40.51 | 62.95 | 64.63 | 0.2001 | 59.41 $\pm 1.97$ | [0.0796, 2.8645] | 62.37 $\pm 0.49$ |
|  | PubMed | 57.84 | 69.09 | 70.13 | 0.2386 | 62.70 $\pm 2.10$ | [0.0826, 2.9727] | 64.78 $\pm 1.00$ |
| GT | Cora | 21.80 | 55.36 | 56.77 | 0.2276 | 54.53 $\pm 1.97$ | [0.0761, 2.7400] | 53.36 $\pm 0.04$ |
|  | CiteSeer | 27.56 | 51.75 | 53.51 | 0.3627 | 43.88 $\pm 2.13$ | [0.0794, 2.8570] | 45.02 $\pm 0.06$ |
|  | PubMed | 39.23 | 70.63 | 72.95 | 0.2084 | 63.93 $\pm 2.15$ | [0.0813, 2.9251] | 65.47 $\pm 2.25$ |

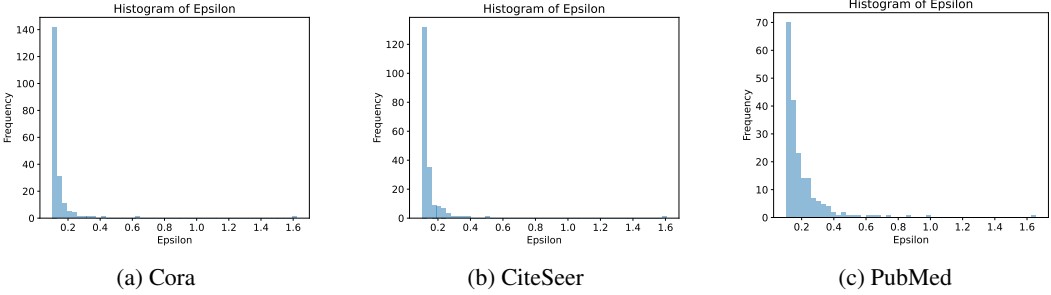

(a) Cora      (b) CiteSeer      (c) PubMed

Figure 9: **Distribution of $\varepsilon$ over teacher ensembles in our DP-GPL+W (DGI, All-in-one, GAT).**

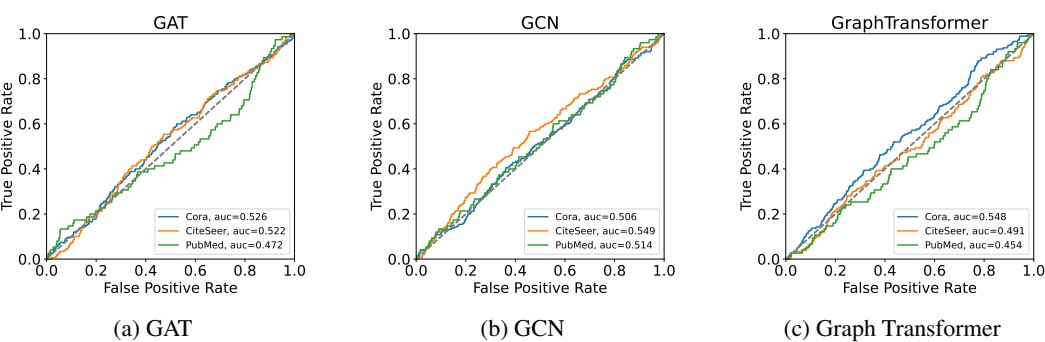

(a) GAT      (b) GCN      (c) Graph Transformer

Figure 10: **AUC-ROC curve of our MIA against DP-GPL (Cora, 5 shots).** Generally, all curves are very close to the dash line (random guess), which shows that DP-GPL is effective against MIA.

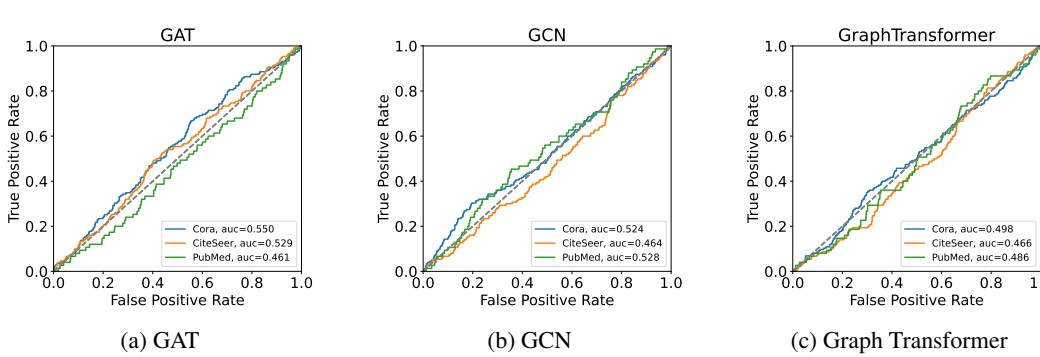

(a) GAT        (b) GCN        (c) Graph Transformer

Figure 11: **AUC-ROC curve of our MIA against `DP-GPL+W` (Cora, 5 shots).** Generally, all curves are very close to the dash line (random guess), which shows that `DP-GPL+W` is also effective against MIA.

