# OpenReview forum: "DP-GPL: Differentially Private Graph Prompt Learning"
_ICLR.cc/2025/Conference — Submitted to ICLR 2025_

### Official Review · Reviewer_61mB · 2024-10-18

**Soundness:** 1
**Presentation:** 2
**Contribution:** 2
**Rating:** 3
**Confidence:** 4

**Summary:**

The objective of this paper is to develop a differentially private (DP) form of graph prompt learning, i.e., adapting a pre-trained graph neural network to a new ("downstream") graph dataset by learning a transformation ("prompt") of the downstream graph, rather than fine-tuning the model's parameters.

To demonstrate the importance of achieving this objective, the authors first showcase the effectiveness of a standard membership inference attack method on graph prompt methods.

As a provable defense against this and other forms of privacy leakage, the authors then propose a method inspired by the well-known PATE framework, in which an ensemble of teacher *models* is first trained on a disjoint partition of sensitive *dataset*, and then used to generate labels for training of a student *model* on public data in a differentially private manner.
Specifically, the authors' method ("DP-DGL") trains an ensemble of teacher *prompts* on a disjoint partition of a sensitive *graph* and then uses it to generate labels for training of a student *prompt* on a public graph. The authors further propose a graph-specific procedure for partitioning the sensitive data and weighting the ensemble based on node centrality ("DP-DGL+W").

They finally evaluate the privacy-accuracy tradeoff of their two methods on three standard citation graph datasets (Cora, CiteSeer, PubMed), comparing their method to (1) DP-SGD prompt learning, (2) direct application of a pre-trained GNN (3), direct use of the teacher prompt ensemble, and (4) direct use of the best-performing prompt from the  ensemble.

**Strengths:**

* The proposed framework is applicable to arbitrary graph prompt learning methods, allowing it to directly benefit from future advances (I would encourage the authors to put more emphasis on this strength in their manuscript).
* First conducting a membership inference attack serves as a convincing motivation for why the considered problem should be worth studying.
* The authors do a good job in contrasting their method to existing DP prompt-learning methods outside the graph domain (including methods inspired by PATE).
* Increasing the utility of PATE via data-dependent ensemble weights is a novel approach that could be of interest for a larger audience.
* The results convincingly demonstrate that the student prompt can achieve comparable utility to the teacher ensemble while offering formal privacy guarantees.
* Results are presented with standard deviations / error bars.
* The general writing style is clear and concise.
* The authors provide an implementation for reproducibility.

**Weaknesses:**

* The manuscript does not specify which neighboring relation / notion of graph privacy is considered. Existing literature on graph DP primarily focuses on either *edge-level* or *node-level* privacy, i.e., preventing privacy leakage due to adjacency changes (see Table 1 in [1]). This adjacency-dependency is the unique challenge that differentiates DP in graphs from DP in other domains.
* The DPG-DGL-W+ procedure proposed in this paper is neither edge-level nor node-level private. Specifically, the partitioning/weighting procedure (Algorithm 2) depends on centrality scores, which depend on the graph adjacency. Since no steps are undertaken to limit its sensitivity and introduce randomness, the leaked adjacency information may be arbitrarily large ($\epsilon=\infty$).
* Due to the previous point, novelty is severely limited. As stated by the authors, the main differentiator from the DP prompt learning method of Duan et al. (2023) [2] is precisely this problematic partitioning/reweighting procedure (see ll.183-187 and compare Fig. 1 from this manuscript to Fig. 1b from [2]).
* Aside from the conceptually interesting idea of data-dependent ensemble weighting, the work introduces no new methods or theoretic results. As such, it will likely be of limited interest for readers outside the "trustworthy graph prompt learning" sub-niche within the (growing and important) "trustworthy graph ML" niche.
* The manuscript does not provide sufficient background information to follow it without consulting prior work. Specifically, the general paradigm of graph prompt learning is only vaguely and indirectly explained by summarizing related work in Section 2.2. I would strongly suggest introducing a "background" section that defines a generic graph prompt learning procedure and explains the involved terminology (how are GNNs "queried", what is a "target prompt", what is a "shot" in the graph context, ...?)
* The related work section does not discuss prior work on DP for graphs, such as DP graph analysis [3], specialized DP-SGD for graphs [4], specialized DP-GNN architectures [5], or PATE for graphs [6] (see [1] for a survey).

**Minor weaknesses**
* The results only specify $\epsilon$, but not $\delta$ of the $(\epsilon,\delta)$-DP prompts (e.g. Table 1).
* The reported accuracies in Tables 1,5,7-11, even without DP guarantees ($\epsilon=\infty$), are very low. One could probably achieve much higher accuracy by discarding the private data and performing standard semi-supervised learning on the public subgraph of Cora/Citeseer/PubMed. This calls into question why one would want to use prompt learning in the first place.
* While the paper demonstrates that membership inference attacks against graph prompt learning are effective, it does not explain why these should be of any concern. A motivating real-life example (e.g. leaking of communication patterns) might be helpful.
* The authors appear to be using the original DP-SGD w/ moments accounting technique of Abadi et al. (2016) as a "naive solution" baseline. Moments accounting is an obsolescent method that can greatly overestimate privacy leakage due to lossy conversion from Renyi-DP to approximate DP. Using a tight numeric accounting method (e.g. [8]) might yield stronger privacy guarantees and invalidate the findings.
* The "naive solution" appears to ignore the privacy leakage caused by message passing in GNNs (see, e.g., [4]) (Taking this into account would actually be beneficial for the authors, since standard DP-SGD underestimates privacy leakage in MPNNs).
* The privacy analysis of DP-GPL+W omits one final step (see ll. 385-387). The overall privacy guarantee should be $(\max\{\epsilon_1,\dots,\epsilon_N\}, \delta)$-DP, because the definition of DP involves worst-case pairs of datasets (i.e., we have to make the pessimistic assumption that the modified data is used to train the least private teacher).
* The pseudo-code in Algorithm 3 is not sufficiently specific about how exactly prompts are trained / used, which makes it hard to reproduce the results from the paper alone. The authors do however provide an implementation in the supplementary material.

[1] Müller et al. Differential Privacy Guarantees for Analytics and Machine Learning on Graphs: A survey of Results. Journal of Privacy and Confidentiality Vol. 14  2023
[2] Duan et al. Flocks of Stochastic Parrots: Differentially Private Prompt Learning for Large Language Models. NeurIPS 2023
[3] Kasiviswanathan et al. Analyzing Graphs with Node Differential Privacy. Theory of Cryptography 2013.
[4] Daigavane et al. Node-Level Differentially Private Graph Neural Networks. arXiv:2111.15521
[5] Sajadmanesh et al. GAP: Differentially Private Graph Neural Networks with Aggregation Perturbation. USENIX 2023
[6] Olatunji. Releasing Graph Neural Networks with Differential Privacy Guarantees. TMLR 06/2023.
[7] Abadi et al. Deep Learning with Differential Privacy. CCS16.
[8] Google DP Team. Privacy Loss Distributions. https://raw.githubusercontent.com/google/differential-privacy/main/common_docs/Privacy_Loss_Distributions.pdf

---

Given these weaknesses, specifically the non-privacy of the proposed DP-GPL-W+ procedure, and the lack of novelty of DP-GPL compared to (Duan et al., 2023) [2], **I recommend rejection of the manuscript**.
I do however believe that private graph prompt learning could be of interest to the trustworthy graph ML community, and would encourage the authors to submit an expanded revision to future conferences (see also suggestions below).

**Questions:**

* Could you please explain l.15 in Algorithm 3 and the blue lines in Fig. 1? That is, how can prompts trained on the private graph be applied to a separate public node/graph? Existing GPL literature (e.g. GPPT) seems to focus on prompting/transforming one specific downstream graph, rather than finding prompts that generalize across graphs.
* How are Cora / Citeseer / Pubmed split into private and public data?
* Is the private part of these graphs used during inference on the public part of the graph?

---

Since there is no other field for general comments, here are a few more suggestions for future revisions (**There is no need to respond to these comments**):

* As can be inferred from my comments above, future revisions would benefit from focusing more on the aspects that are unique to graph prompt learning (as opposed to language / vision / ... -prompt learning)
* One way of doing this would be by focusing on adjacency changes / node insertions
* Another way of doing this could be looking into how message-passing interacts with graph prompting (i.e. how is private information propagated through the graph when training a prompt)
* You might also want to consider the effect of message passing when prompting at inference time (unless the private and public part of the graph are kept disjoint)
* I would suggest highlighting best/second-best accuracies in all tables to improve readability.

---

> ### Author Response · Authors · 2024-11-27
>
> >**The manuscript does not specify which neighboring relation / notion of graph privacy is considered.**
>
> We thank the reviewer for pointing that out. Our paper considers node-level DP, i.e., we aim to protect the privacy of the nodes used in the graph prompt training. We clarified this information in the introduction, related work, and method section of our updated paper.
>
> >**The DPG-DGL-W+ procedure proposed in this paper is neither edge-level nor node-level private. Specifically, the partitioning/weighting procedure (Algorithm 2) depends on centrality scores, which depend on the graph adjacency.**
>
> We are happy to clarify our setup further to address this comment. We see the reviewer’s concern that the weight of a teacher will give away the connectivity of the training nodes, which could be considered as private information that is not supposed to leak. However, in the similar vein to standard PATE, we assume that the teachers and the training procedure of the student are kept secret, i.e., they are not visible to the outside world. The only part that is visible to the outside world is the student prompt. Thereby, the teacher weights are protected.
>
> We have extended the paper by this explanation.
>
> >**The manuscript does not provide sufficient background information to follow it without consulting prior work. Specifically, the general paradigm of graph prompt learning is only vaguely and indirectly explained by summarizing related work in Section 2.2. I would strongly suggest introducing a "background" section that defines a generic graph prompt learning procedure and explains the involved terminology (how are GNNs "queried", what is a "target prompt", what is a "shot" in the graph context, ...?)**
>
> We thank the reviewer for the suggestion. We have extended the "background" section in the revised manuscript, see Section 2.2.
>
> >**The related work section does not discuss prior work on DP for graphs, such as DP graph analysis [3], specialized DP-SGD for graphs [4], specialized DP-GNN architectures [5], or PATE for graphs [6] (see [1] for a survey).**
>
> We thank the reviewer for the suggestion. We have added a discussion of prior work on DP for graphs in the related work section in the revised manuscript, see Section 2.4.
>
> >**The results only specify ϵ, but not δ of the (ϵ,δ)-DP prompts (e.g. Table 1).**
>
> We have updated the tables accordingly.
>
> >**The reported accuracies in Tables 1,5,7-11, even without DP guarantees (ϵ=∞), are very low. One could probably achieve much higher accuracy by discarding the private data and performing standard semi-supervised learning on the public subgraph of Cora/Citeseer/PubMed. This calls into question why one would want to use prompt learning in the first place.**
>
> The low performance observed stems from the fact that we only operate in the 5-shot setting. However, not every private task can be trained based on some public equivalent. Using graph prompting allows us to adapt the GNNs to private tasks that differ from publicly available tasks. For instance, we can leverage graph prompt learning to effectively and efficiently adapt a pre-trained GNN model, which is pre-trained on edge prediction tasks, to a node classification downstream task.
>
> >**The overall privacy guarantee should be (maxϵ1,…,ϵN,δ)-DP, because the definition of DP involves worst-case pairs of datasets (i.e., we have to make the pessimistic assumption that the modified data is used to train the least private teacher).**
>
> In our paper, we presented the (min_ϵ, max_ϵ)-DP guarantee of DP-GPL+W. This is in line with the reporting of privacy guarantees under heterogenous DP [A] which reports (min_ϵ, …,  max_ϵ), showing the epsilon for every privacy group. We shortened it to (min_ϵ, max_ϵ)-DP. In the updated version of the paper, we have changed to reporting (max_ϵ, δ)-DP, but for completeness, provided table 12 in Appendix A.4.4 where we report (min_ϵ, …,  max_ϵ)-HDP.
>
> **References**
>
> [A] Mohammad Alaggan, Se ́bastien Gambs, and Anne-Marie Kermarrec. Heterogeneous differential privacy. Journal of Privacy and Confidentiality, 7(2), 2016.

---

> > ### Author Response · Authors · 2024-11-27
> >
> > >**The pseudo-code in Algorithm 3 is not sufficiently specific about how exactly prompts are trained / used.**
> >
> > We thank the reviewer for the suggestion. We have provided a detailed formulation of the graph prompt in Section 2.2 in the revised manuscript.
> >
> > >**That is, how can prompts trained on the private graph be applied to a separate public node/graph?**
> >
> > In our setup, the public nodes are selected from the same dataset as the training/testing nodes so that the prompts trained on the private graph can be applied to a separate public node.
> > Taking the GPF-plus as an example, the prompt is a trainable vector that has the same dimension as the node feature. If we select the public nodes from the same dataset as the training/testing nodes, the trained prompts can be adapted to the public node by adding the learned vector to the node feature. Otherwise, the trained prompts can not be applied to a public node as the dimension of the prompt may be different from that of the original node feature.
> >
> > >**How are Cora / Citeseer / Pubmed split into private and public data?**
> >
> > We randomly select 50% of the nodes as the private data and the remaining 50\% as the public data. Within the public data, we randomly select 50 nodes as the query nodes and the remaining nodes as the testing data. We have added this information in the revised manuscript. Upon splitting, we didn’t consider the edges between the nodes, i.e., we consider each node an independent data point.
> >
> > >**Is the private part of these graphs used during inference on the public part of the graph?**
> >
> > In our case, the private part of the graph is not used during inference on the public part of the graph.

---

> ### Comment · Reviewer_61mB · 2024-11-28
>
> Thank you, your rebuttal addresses many of my listed concerns.
>
> However, my main concern, i.e., non-privacy of DP-DGL+W remains.
> To clarify, the issue arises even when the teacher prompts are obfuscated.
>
> The GNNMax privacy analysis in the paragraph "Noisy Teacher Vote Aggregation" of Section 4.2 assumes that the teacher weights $w_i$ are constant. It only accounts for the sensitivity of the teacher votes $y_i$.
> However, due to being a function of centrality scores, they are data-dependent and we may have $w_i(A) \neq w_i(A')$ for two adjacent adjacency matrices $A \simeq A'$. A valid privacy analysis would also need to take into acount the sensitivity of $w_i(\cdot)$ and the resultant sensitivity of the aggregated teacher vote.
>
> There is a chance I might be missing something. If your method accounts for this data-dependence, feel free to point me to the corresponding section in your paper or the PATE paper.

---

> > ### Author Response · Authors · 2024-11-28
> >
> > We thank the reviewer for raising these important points regarding the privacy analysis of DP-GPL+W.
> >
> > We have conducted a detailed privacy analysis of DP-GPL+W in Section 4.3.
> > Specifically, as stated in Proposition 1, the sensitivity of the teacher is equal to the weight of the teacher.

---

> ### Comment · Reviewer_61mB · 2024-11-29
>
> Let me maybe rephrase differently: Your analysis assumes that there is a single, constant "weight" $w_i$ per teacher. Instead we have different weights $w_i(A)$ and $w_i(A')$ when consering any two adjacent graphs.
> Thus, the heterogeneous per-teacher sensitivity, which should be data-independent for GNNMax to provide valid guarantees, is ill-defined.
>
> This is the same issue that is discussed in literature with local sensitivity analysis (propose test release / smooth local sensitivity).
>
> Also, without further analysis of how the centrality scores behave, we have to make the worst-case assumption that we have $w_i(A)=0$, $w_j(A) =1$ for the original graph and $w_i(A')=1$, $w_j(A')=0$ for the modified graph with some $i \neq j$, which means that the aggregate prediction can change arbitrarily.
>
> Disclaimer: I will likely not respond to additional comments on this issue, unless you actually resolve some fundamental misunderstanding on my end.

---

> > ### Author Response · Authors · 2024-12-01
> >
> > We thank the reviewer for their valuable insights, and we will revise the paper accordingly for our next submission.

---

### Official Review · Reviewer_tCRM · 2024-11-03

**Soundness:** 2
**Presentation:** 3
**Contribution:** 2
**Rating:** 5
**Confidence:** 3

**Summary:**

The paper combines the PATE framework with differential privacy to provide robust privacy protection in graph prompt learning while maintaining high utility. The topic is new and interesting.

**Strengths:**

1. The paper is well-structured, easy to read, and well-written.
2. It addresses privacy risks in graph prompt learning and proposes two non-gradient-based methods.

**Weaknesses:**

1. Lack of Experimentation: In Section 3, the author demonstrates through MIA experiments that prompts based on private graphs can lead to node privacy leakage, proving the existence of privacy risks and subsequently proposing a DP-based protection method. However, while DP provides theoretical guarantees, it is unclear whether there is any impact on the effectiveness of attacks after DP is applied. The current experiment section lacks a demonstration of MIA attack results after DP has been implemented.
2. Unclear Theoretical Section: Graph DP is generally divided into node-DP and edge-DP. The target of the author’s attack experiments is to determine whether a certain node is in the prompt's training graph dataset, which falls under node-DP. However, node-DP involves protecting node features, edges, and labels. In the proposed method, there does not appear to be any protection for labels.

**Questions:**

1. How effective are DP-GPL and DP-GPL+W in resisting MIA? Could the author conduct and report MIA experiments on the DP-protected models (DP-GPL and DP-GPL+W) and compare the results to the unprotected baselines?
2. In DP-GPL+W, the teacher voting is weighted based on node centrality, and the partitioning strategy changes to outperform DP-GPL. Why is this effective? Could the author provide experimental or theoretical evidence, such as ablation studies?
3. Is the proposed method only applicable to homogeneous graphs, or is it suitable for heterogeneous graphs as well?
4. The current experiments focus only on the k-shot scenario where k=5. Could performance comparisons for smaller k-values be added?
5. According to weakness 2, please clarify how your method addresses label privacy.

---

> ### Author Response · Authors · 2024-11-27
>
> >**The current experiment section lacks a demonstration of MIA attack results after DP has been implemented.**
>
> We have conducted additional MIA experiments on the DP-protected models (DP-GPL and DP-GPL+W). Specifically, we also draw the AUC-ROC curves to evaluate the performance of the MIA against the DP-protected models. The results show that all curves are very close to random guess, which shows that our proposed methods are effective against MIA. For example, on Cora and GAT, the AUC score is 0.703 before applying our methods, while that is 0.526 and 0.550 after applying DP-GPL and DP-GPL+W, respectively.
> We have added the results in the revised manuscript, see Appendix A.4.8.
>
> >**In the proposed method, there does not appear to be any protection for labels.**
>
> We adapted the paper to clarify our privacy setup. We aim to provide node-level DP. This aims at protecting the nodes, however, does not provide label privacy. Hence, providing label privacy is out of the scope of this work.
>
> >**In DP-GPL+W, the teacher voting is weighted based on node centrality, and the partitioning strategy changes to outperform DP-GPL. Why is this effective? Could the author provide experimental or theoretical evidence, such as ablation studies?**
>
> We have conducted experiments to show the effectiveness of DP-GPL+W compared to DP-GPL. Specifically, we have compared the performance of DP-GPL and DP-GPL+W on three standard citation graph datasets (Cora, CiteSeer, PubMed), as shown in Table 1, 7-11 in the paper. The results show that DP-GPL+W outperforms DP-GPL in terms of privacy-utility trade-offs.
>
> >**Is the proposed method only applicable to homogeneous graphs, or is it suitable for heterogeneous graphs as well?**
>
> There is no restriction on the graph data in our proposed method. As long as the teacher prompts can be trained on the private data, our method can output a private graph prompt for the downstream task. Taking the heterogeneous ogbn-mag network [A] from the OGB dataset suit as an example, we can first split the private nodes into different partitions where each partition contains nodes of different types, then train the corresponding teacher prompt on each partition.
>
> >**The current experiments focus only on the k-shot scenario where k=5. Could performance comparisons for smaller k-values be added?**
>
> We thank the reviewer for this suggestion. We will include the experiments with smaller k-values in the revised manuscript.
>
> >**Please clarify how your method addresses label privacy.**
>
> We aim to protect the privacy of the nodes in the prompt's training data, however, providing label privacy is not in scope. We have clarified this in the revised manuscript.
>
> **References:**
>
> [A] Kuansan Wang, Zhihong Shen, Chiyuan Huang, Chieh-Han Wu, Yuxiao Dong, and Anshul Kanakia. Microsoft academic graph: When experts are not enough. Quantitative Science Studies, 1(1):396–413, 2020.

---

### Official Review · Reviewer_Jp8Y · 2024-11-03

**Soundness:** 2
**Presentation:** 3
**Contribution:** 2
**Rating:** 3
**Confidence:** 5

**Summary:**

The paper investigates privacy risks in graph prompts by instantiating a membership inference attack that reveals significant
privacy leakage. Then they find that the standard privacy method, DP-SGD, fails to
provide practical privacy-utility trade-offs in graph prompt learning, likely due to
the small number of sensitive data points used to learn the prompts. As a solution,
they propose two algorithms, DP-GPL and DP-GPL+W, for differentially private
graph prompt learning based on the PATE framework, that generate a graph prompt
with differential privacy guarantees.

**Strengths:**

1. It is the first to show that private information can leak from graph prompts, in particular when
the prompts are tuned over a small number of data points.
2. It shows that naively integrating the DP-SGD algorithms into graph prompt learning yields
impractical privacy-utility trade-offs.
3. It proposes DP-GPL and DP-GPL+W, two algorithms based on the PATE framework
to implement differential privacy guarantees into graph prompt learning.
4. Their new methods achieve both
high utility and strong privacy protections over various setups.

**Weaknesses:**

The privacy mechanism seems like is a direction adaptation from PATE. I believe at least the core part (voting and then adding noise) is very similar to PATE. The other part, despite being claimed by the author to be different from PATE, I do not find it interesting because this is GNN task, which is of-course different from the original task of PATE.

**Questions:**

Should it be better to bring algorithm 3 into the main body of this paper?
For the MIA experiment, LiRA attach originally reports true positive at low false positive, which is more sensible than the AUC metric. Have the author considered this?

---

> ### Author Response · Authors · 2024-11-27
>
> >**Should it be better to bring algorithm 3 into the main body of this paper?**
>
> We thank the reviewer for the suggestion. We didn’t bring algorithm 3 into the main body due to the page length of the paper.
>
> >**For the MIA experiment, LiRA attach originally reports true positive at low false positive, which is more sensible than the AUC metric. Have the author considered this?**
>
> We thank the reviewer for the suggestion. Instead of evaluating the worst case of MIA, i.e., true positive at low false positive rate, we aim to evaluate the overall performance of the MIA against graph prompt learning. Thus, we consider the AUC-ROC curves to evaluate the performance of the MIA, following the same evaluation metric as in [A] (Figure 2-b).
>
> **References:**
>
> [A] *”Flocks of Stochastic Parrots: Differentially Private Prompt Learning for Large Language Models”*. Haonan Duan, Adam Dziedzic, Nicolas Papernot, Franziska Boenisch, NeurIPS 2023.

---

> > ### Comment · Reviewer_Jp8Y · 2024-12-02
> > **Response**
> >
> > Thanks for the response. However, the authors did not respond to my concerns about novelty and comparison with PATE or previous DP-ICL. I will maintain my score

---

> > > ### Author Response · Authors · 2024-12-03
> > >
> > > We thank the reviewer for taking the time to review our work.
> > >
> > > >**Novelty concerns**
> > >
> > > We would like to emphasize that, to the best of our knowledge, _no prior work_ has explored membership inference threats in the context of graph prompt learning. Our study is the first to investigate the privacy risks associated with graph prompt learning and propose two PATE-based methods, i.e., DP-GPL and DP-GPL+W, to implement differential privacy guarantees into graph prompt learning.
> > >
> > > Notably, DP-GPL+W leverages the inherent structure of the graph data to improve the privacy-utility trade-offs, distinguishing our approach from the original PATE. Moreover, unlike the original PATE, which focuses on privately training _models_, our methods aim to privately train _graph prompts_. We will highlight our novelty and contributions in the revised manuscript.

---

### Official Review · Reviewer_xzGh · 2024-11-03

**Soundness:** 3
**Presentation:** 3
**Contribution:** 3
**Rating:** 3
**Confidence:** 4

**Summary:**

The paper probes into the problem of privacy concerning graph prompt learning. The paper first demonstrates the the privacy leakage caused by graph prompt learning via MIA. Secondly, in order to resolve the privacy concern, the paper proposed DP-GPL along with its variant version DP-GPL+W which utilizes the well-known PATE structure for differential private guarantees. The experimental results illustrate that the proposed method can preserve the utility of downstream task while maintaining the privacy of the given data.

**Strengths:**

1. This paper is one of the few papers that investigate the privacy leakage problem in the setting of graph prompt learning. Besides measuring the privacy leakage via MIA to infer the problem of privatized graph prompt learning, the paper also proposes methods to properly alleviate such issues.
2. The experiments is extensive as it covers various datasets along with pre-training methods, rendering the evaluation results holistic.

**Weaknesses:**

1. The differential privacy guarantee mentioned in the paper only concerns DP and RDP, which is particularly set for traditional database and datasets that are i.i.d., i.e. with no correlations. It would be great if the authors could further clarify the DP settings used in the paper, e.g., node or edge differential privacy are considered.

2. Furthermore, the partition scenario that utilize centrality score in DP-GPL+W doesn't seem to be satisfying the partition scenario in the original PATE paper [1]. Particularly, the partition seems to have utilize the private data information without paying any budget in advance which seems to have broken the standard DP guarantee. Although the original PATE allows the partitions to be formed in a non-randomly, it adds to the description that it has to be *naturally and non-randomly* distributed. Namely, the partition could be non-randomly formed by either privacy or regulation concerns that are *a priori* to the actual execution of the training algorithms.

3. Lastly, as DP-GPL+W utilizes the heterogenous definition of DP which are mentioned in [2, 3], the application scenario should be elaborated more on the use case of DP-GPL+W. Particularly, individualized DP refers to privacy standards that could cope with different levels of privacy that are requested by users, i.e., the data holders. However, it seems that in DP-GPL+W, this standard is brought up simply due to the need of weighting mechanism in DP-GPL+W that concerns the graph data itself instead of the different levels of privacy each user or node requested. Does that implies that nodes with higher centrality score are naturally giving out more information in DP-GPL+W regardless of the users' request in privacy levels? Also, how does this associates with the node or edge definition privacy mentioned in the first point?


[1]: Nicolas Papernot, Martín Abadi, Úlfar Erlingsson, Ian Goodfellow, and Kunal Talwar. Semisupervised knowledge transfer for deep learning from private training data. In Proceedings of the 5th International Conference on Learning Representations (ICLR), 2017.\
[2]: Boenisch, Franziska, et al. "Individualized PATE: Differentially private machine learning with individual privacy guarantees." arXiv preprint arXiv:2202.10517 (2022).\
[3]:Alaggan, Mohammad, Sébastien Gambs, and Anne-Marie Kermarrec. "Heterogeneous differential privacy." arXiv preprint arXiv:1504.06998 (2015).

**Questions:**

1. As mentioned in the first and three point of weaknesses, it would be great the authors could specifically explain the definition of graph DP used in the paper. On the other hand, the authors are also welcomed to discuss the facet mentioned in the third point. Particularly, what would a user be faced with if his/her data possesses high centrality score but requested a low or high privacy level in the individual DP settings?

2. Similar to the reason raised in second point, it would be great if the authors could cook up a specific proof tailored to DP-GPL+W as there are steps concerning private data being added into the original PATE pipeline.

3. Regarding the description of [1] in Section 2.5, I believe that the setting of label mapping is natural in image domain the fact that model reprogramming (MR) / visual prompting (VP) aims to re-use existing foundational models for either domain transfer or further fine-tuning. In that way, this doesn't seem to be a comparable advantage over GPL as it is simply different. I would suggest the authors to revise the part in the paper.




[1]: Yizhe Li, Yu-Lin Tsai, Chia-Mu Yu, Pin-Yu Chen, and Xuebin Ren. Exploring the benefits of visual prompting in differential privacy. In Proceedings of the IEEE/CVF International Conference on Computer Vision, pp. 5158–5167, 2023.

---

> ### Author Response · Authors · 2024-11-27
>
> >**It would be great if the authors could further clarify the DP settings used in the paper, e.g., node or edge differential privacy are considered.**
>
> Considering that the goal of graph-level DP is to protect the property of the graph, e.g., node degree, clustering coefficient, community structure, etc., and the goal of node-level DP is to protect the privacy of the nodes in the graph, e.g., node features, edges, and labels, we here focus on node-level DP as we aim to protect the privacy of the nodes used in the graph prompt training.
>
> Also, note that there is *no connection* between the training nodes of the different teacher prompts, so the classical PATE privacy guarantee is still applicable in our case. We have added this clarification in the revised manuscript.
>
> >**The partition scenario that utilize centrality score in DP-GPL+W doesn't seem to be satisfying the partition scenario in the original PATE paper [1].**
>
> In the original PATE paper, the partition could be non-randomly formed by either privacy or regulation concerns that are a priori to the actual execution of the training algorithms. However, in our paper, we first assign different weights to partitions according to the average centrality score of the private subset and then assign the same privacy budget for each teacher, i.e., ϵ=2 in our paper. Finally, the actual privacy cost for each partition is calculated following Algorithm 2 in [2].
>
> >**Furthermore, the partition scenario that utilize centrality score in DP-GPL+W doesn't seem to be satisfying the partition scenario in the original PATE paper [1].**
>
> PATE can use arbitrary partitioning, the only requirement is that in the standard PATE the partitions are truly mathematical structures, so there is no data overlap between the partitions. For example, arbitrary partitioning is used in the CaPC framework [A] that leverages PATE and does not incur additional privacy costs. In Private kNN [B], based on PATE, every data point is a separate teacher. The selection of the data points to teachers is also specified explicitly in PATE [1]. Thus, the teachers can be created arbitrarily.
>
> **References:**
>
> [A] *”CaPC Learning: Confidential and Private Collaborative Learning”*  Christopher A. Choquette-Choo, Natalie Dullerud, Adam Dziedzic, Yunxiang Zhang, Somesh Jha, Nicolas Papernot, Xiao Wang, ICLR 2021.
>
> [B]  *”Private-kNN: Practical Differential Privacy for Computer Vision”* Yuqing Zhu, Xiang Yu, Manmohan Chandraker, Yu-Xiang Wang, CVPR 2020.

---

> > ### Author Response · Authors · 2024-11-27
> >
> > >**Does that implies that nodes with higher centrality score are naturally giving out more information in DP-GPL+W regardless of the users' request in privacy levels? Also, how does this associates with the node or edge definition privacy mentioned in the first point?**
> >
> > Note that while we use methods from heterogeneous DP, we do not operate in a heterogeneous DP setup, i.e., we do not assume that any node has different privacy requirements than any other node. We set the privacy budget to $\varepsilon=2$ for every node and make sure that no node exceeds this budget. In effect, it is true that non-central nodes usually get even higher protection than that, but DP provides a worst-case guarantee; hence, $\varepsilon=2$ still holds for every node.
> >
> > Further, we want to clarify that there is only one user in our case, i.e., the owner of the private data, who aims to learn a private graph prompt based on the private data. We assume that nodes with higher centrality scores contain more information, which can contribute more to the learning of the graph prompt. To leverage this, we assign higher weights to partitions with higher centrality scores.
> > Importantly, while we assign the same privacy budget for each partition, the actual privacy cost for each partition is calculated based on the weights. This strategy improves privacy-utility trade-offs further.
> >
> > Together with the additional general discussion on the DP specific to the graph domain, we have also provided a detailed explanation of the node-level DP in our methods in the revised manuscript.
> >
> > >**What would a user be faced with if his/her data possesses high centrality score but requested a low or high privacy level in the individual DP settings?**
> >
> > Again, we do not assume any heterogeneous privacy preferences. In our case, we assign the same privacy level to all nodes, i.e., teacher prompts. To achieve this goal, we propose the DP-GPL+W method, which assigns different weights to partitions based on the centrality scores of the nodes in the private subset. Different weights lead to different privacy costs for each partition, but the maximum privacy budget is the same for all partitions.
> >
> > >**It would be great if the authors could cook up a specific proof tailored to DP-GPL+W as there are steps concerning private data being added into the original PATE pipeline.**
> >
> > We thank the reviewer for the suggestion. We have updated the PATE pipeline in the revised manuscript, e.g., Figure 1, to clarify the steps specifically concerning private graph data being added to the original PATE pipeline. Overall, as the individual teacher prompts contain non-connected subgraphs, the privacy analysis is not affected.

---

### Meta-Review · Area_Chair_aeEH · 2024-12-17

**Metareview:**

This paper proposes DP-GPL and DP-GPL+W that combine the PATE framework for differentially private graph prompt learning. Reviewers mention that this is an interesting direction in the "trustworthy graph prompt learning" sub-community that is also of practical interest.

However, this paper does not meet the bar for ICLR. Here are some of the reasons.

1) There were strong concerns about novelty and comparison with PATE or previous the DP-ICL.
2) The DPG-DGL-W+ procedure proposed in this paper is neither edge-level nor node-level private. This limits the novelty of the paper.
3) No new methods and theoretical results are proposed.

**Additional Comments On Reviewer Discussion:**

There were some discussions between the authors and reviewers. However, the reviewers were generally still unconvinced by the authors' responses. The authors themselves acknowledge the weaknesses, which have been clearly spelled out by the reviewers. They have resolved to tae these into account in a future submission.

---

### Decision · Program_Chairs · 2025-01-22

Reject